# A multi-omic characterization of temperature stress in a halotolerant *Scenedesmus* strain for algal biotechnology

Sara Calhoun [1,2,7], Tisza Ann Szeremy Bell [3,4,7], Lukas R. Dahlin [5,7], Yuliya Kunde[3], Kurt LaButti [1], Katherine B. Louie [1], Andrea Kuftin[1], Daniel Treen [1], David Dilworth [1], Sirma Mihaltcheva[1], Christopher Daum [1], Benjamin P. Bowen [1], Trent R. Northen [1], Michael T. Guarnieri [5], Shawn R. Starkenburg [3✉] & Igor V. Grigoriev [1,2,6✉]

Microalgae efficiently convert sunlight into lipids and carbohydrates, offering bio-based alternatives for energy and chemical production. Improving algal productivity and robustness against abiotic stress requires a systems level characterization enabled by functional genomics. Here, we characterize a halotolerant microalga *Scenedesmus* sp. NREL 46B-D3 demonstrating peak growth near 25 °C that reaches 30 g/m$^2$/day and the highest biomass accumulation capacity post cell division reported to date for a halotolerant strain. Functional genomics analysis revealed that genes involved in lipid production, ion channels and antiporters are expanded and expressed. Exposure to temperature stress shifts fatty acid metabolism and increases amino acids synthesis. Co-expression analysis shows that many fatty acid biosynthesis genes are overexpressed with specific transcription factors under cold stress. These and other genes involved in the metabolic and regulatory response to temperature stress can be further explored for strain improvement.

[1] US Department of Energy Joint Genome Institute, Lawrence Berkeley National Laboratory, Berkeley, CA, USA. [2] Environmental Genomics and Systems Biology Division, Lawrence Berkeley National Laboratory, Berkeley, CA, USA. [3] Applied Genomics Team, Bioscience Division, Los Alamos National Laboratory, Los Alamos, NM, USA. [4] Division of Biological Sciences, Genome Core, University of Montana, Missoula, MT, USA. [5] National Bioenergy Center, National Renewable Energy Laboratory, Golden, CO, USA. [6] Department of Plant and Microbial Biology, University of California Berkeley, Berkeley, CA, USA. [7] These authors contributed equally: Sara Calhoun, Tisza Ann Szeremy Bell, Lukas R. Dahlin. ✉email: shawns@lanl.gov; ivgrigoriev@lbl.gov

Microalgae are promising renewable biomass resources and phototrophic biocatalysts for the production of fuel and chemical intermediates[1]. However, biotechnological deployment of microalgae faces numerous economic and sustainability hurdles. With an estimated 1 million species[2], algae are diverse in habitat and metabolic output yet the individual identification, characterization, and development of more resilient and productive strains is essential to achieving economically viable mass cultivation. While bioprospecting and strain screening have proven to be an effective approach to acquiring increasingly better performing taxa[3–5], the time and resources required to assess growth physiology, metabolite production, and robustness to environmental stressors create a bottleneck. To this end, we have recently established a screening and characterization pipeline using simulated outdoor deployment screening methodology, as described previously[3,4]. Top-candidates emerging from this pipeline are evaluated for growth rate, biomass accumulation, halotolerance, and carbon storage capacity. Thus, the primary goals of this study were to first identify and characterize a high-productivity strain of interest, then subject it to a battery of multi-omics analyses to baseline its metabolic response to known environmental pressures to identify genetic targets and/or biochemical pathways that when modified, may enhance the stability and productivity of the algae in suboptimal cultivation conditions.

One of the primary environmental drivers of algal productivity is temperature. During mass cultivation in open ponds, fluctuations in temperature can stress or temporarily distort metabolic network processes affecting growth, overall productivity, and biomass composition[6–8]. As reported for several taxa[9,10], cold stress, either individually[11,12] or in some combination with another environmental stress[13], reduces growth/productivity and can result in increased lipid that exceeds the effects of nitrogen starvation[14] including unsaturated fatty acids[11,12,15] that maintain membrane permeability and fluidity[15,16]. However, the underlying mechanisms that control and regulate these and other physiological responses and adaptation to temperature stress are largely unknown. A comprehensive, systems level analysis of promising strains grown under industrially relevant suboptimal temperatures can help identify genetic targets and biochemical pathways that when modified, enhance stability and productivity of the algae during outdoor cultivation.

Here, we report on the initial screening and detailed characterization of the metabolic response of *Scenedesmus* sp. NREL 46B-D3 to temperature stress. *Scenedesmus* sp. NREL 46B-D3 is a high-productivity halotolerant microalga with exemplary mass cultivation potential. Following cessation of cell division (stationary phase), this strain shows remarkable photosynthetic activity, concurrently accumulating new biomass almost exclusively as lipids and carbohydrates. This phenotype provides an intriguing contrast to another top-candidate strain emerging from our pipeline, *Picochlorum renovo*, which lacks biomass accumulation capacity post cell division[3]. We sequenced the genome and profiled the transcriptomic and metabolomic response of this strain under high and low-temperature stress conditions. Metabolomic analysis showed increased levels of certain triacylglycerol species and shifts in the relative abundances of membrane lipids. Differential expression analysis of transcriptomic data allowed identification of unique and conserved transcription factors, some of which were associated with lipid biosynthesis. The resultant data lays the foundation for subsequent genetic manipulation, and strain enhancement[17–20] to improve robust outdoor cultivation at suboptimal environmental temperatures.

## Results

**Strain characterization under summer conditions**. Using a custom built photobioreactor, 107 halotolerant algal strains were screened under simulated summer growth conditions (temperature and PAR cycling), as described in Dahlin et al.[4]. Briefly, 100 mL cultures were screened under the following conditions: 21–32 °C temperature cycling, 35 g/L salinity, 0 to 965 µmol/m²/s light cycling, and constant 2% $CO_2$ sparging[3]. Of the algae screened, *Scenedesmus* sp. NREL 46B-D3 showed the highest endpoint biomass accumulation, nearly double the biomass density of the reference strain, *Nannochloropsis salina* CCMP 1776 (Fig. 1b and Supplementary Data File 1), previously identified as a genus with high potential for outdoor cultivation[17,21].

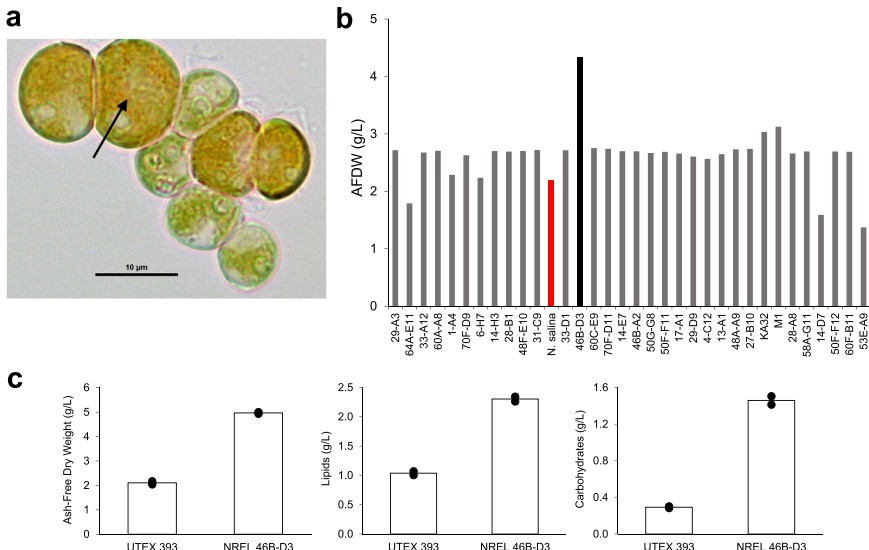

**Fig. 1 Screening of *Scenedesmus* sp. NREL 46B-D3 and culture collection data. a** Micrograph of alga *Scenedesmus* sp. NREL 46B-D3, black arrow indicates observed orange/red pigmentation throughout the cell. **b** Representative culture collection growth screening data. The high biomass accumulation capacity phenotype of *Scenedesmus* sp. NREL 46B-D3 following 6 days of growth is highlighted in black. *Nannochloropsis salina* CCMP 1776 is bolded in red for reference. **c** Endpoint growth comparison of UTEX 393 and NREL 46B-D3 following 9 days of growth-data presented is the average (bar) of two biological replicates (dots).

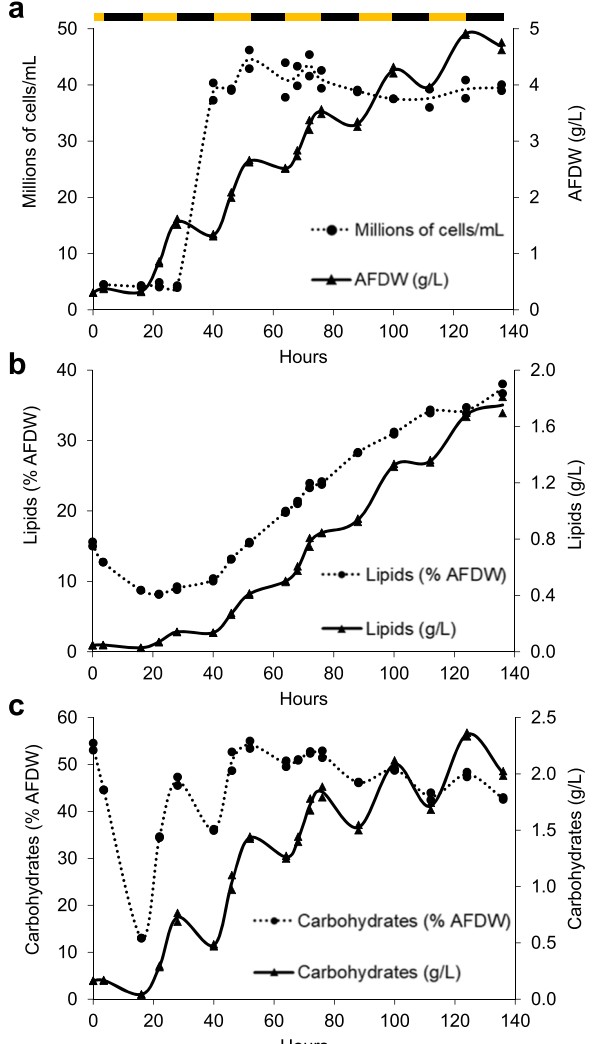

**Fig. 2 Growth and compositional analysis of *Scenedesmus* sp. NREL 46B-D3. a** Growth curves in terms of ash free dry weight (AFDW) and cells per milliliter. **b** Growth curves in terms of lipids as fatty acid methyl esters (g/L) and lipids (% AFDW). **c** Growth curves in terms of carbohydrates (g/L) and carbohydrates (% AFDW). Data points represent the average and standard deviation of *n* = 2 biological replicates. Alternating black and yellow bar represents the lighting cycle (yellow = light cycle, black = dark cycle).

In order to assess the unique *Scenedesmus* sp. NREL 46B-D3 phenotype, we monitored growth under varying temperatures, salinities, and nitrogen:phosphorus (N:P) ratios to further characterize optimal and limiting growth conditions. The maximal growth rate of NREL 46B-D3 was observed at 25 °C, and the highest biomass titer was observed at 17.5 g/L salinity, the lowest salinity tested (Supplementary Fig. 1b). Although increasing salinity resulted in lower biomass accumulation, NREL 46B-D3 demonstrated growth capacity at high salt concentrations, exhibiting growth at 57.5 g/L salinity (equivalent to 1.6X seawater). Growth at differing N:P ratios was most robust at a N:P ratio of 16:0.62, considerably lower than the standard 16:1 Redfield ratio employed for mass algal cultivation (Supplementary Fig. 1c)[22]. A detailed diel batch growth analysis showed synchronized cell division during the dark period. Maximal biomass productivity was 30 g/m²/day with declines in productivity post cell division to 16 g/m²/day. However, 90–98% of biomass accumulated post cell division was in the form of lipids

and carbohydrates (Fig. 2a–c and Supplementary Data File 1). Biomass compositional analysis showed 37% lipids and 43% carbohydrates at the final time point tested (hour 136) (Fig. 2a–c). Putative lipid storage under nitrogen deprivation was predominately in the form of C16:0 and C18:1n9 (Supplementary Fig. 2). After 136 h of growth, glucose and mannose represented the major monomeric sugars following acid hydrolysis, constituting 29% and 14% of the biomass, respectively (Supplementary Fig. 2). To provide context to a previously characterized strain with a similar phenotype, we performed a direct comparison to *Scenedesmus obliquus* UTEX 393, also reported as *Tetradesmus obliquus* or *Acutodesmus obliquus*[23–25], which is currently under evaluation as a top-candidate deployment strain in outdoor field trials. Under the conditions tested here, *Scenedesmus* sp. NREL 46B-D3 accumulated 2.4X biomass, 2.3X lipids, and 5X the amount of carbohydrates following nine days of cultivation (Fig. 1c and Supplementary Data File 1).

**Genome analysis**. The 151.90 Mbp *Scenedesmus* sp. NREL 46B-D3 genome was sequenced using long reads and assembled in 2,661 contigs. 96% of NREL 46B-D3 RNASeq reads were mapped to the genome assembly. The completeness of the predicted protein-coding gene set was estimated to be 93.1% complete based on Chlorophyta BUSCO families (chlorophyta_odb10; 11-20-19)[26,27] (Supplementary Fig. 3). Phylogenetic analysis using the 18S sequences confirmed placement of the strain NREL 46B-D3 in a clade with *Scenedesmus dissociatus* and *Scenedesmus rubescens* amidst other *Scenedesmus* species (Supplementary Fig. 4). The phylogenetic tree based on single-copy orthologous genes places the NREL 46B-D3 genome basal to *Scenedesmus obliquus*, within Sphaeropleales, along with *Monoraphidium neglectum*, *Raphidocelis subcapitata*, and *Chromochloris zofingiensis* (Fig. 3a and Supplementary Data File 2). The genome is likely diploid, as there is separation of alleles, with 1,606 alternate haplotype contigs and an approximate haploid assembly size of 103.4 Mbp. The predicted 17,399 genes in the primary haplotype are comparable in number to other Sphaeropleales genomes but fewer than in the *Scenedesmus obliquus* strains (19,873 in UTEX 3031 haplotype[28] and 19,723 in UTEX 393[24,29]) (Fig. 3b and Supplementary Table 1).

Comparative analysis of the NREL 46B-D3 strain with 11 other Chlorophyta genomes (*Chlamydomonas reinhardtii*, *Chlorella variabilis* NC64A, *Chromochloris zofingiensis*, *Coccomyxa* sp. C-169, *Dunaliella salina*, *Gonium pectorale*, *Monoraphidium neglectum*, *Raphidocelis subcapitata*, *Scenedesmus obliquus* UTEX 3031, *Scenedesmus obliquus* UTEX 393 and *Volvox carteri*) was performed to identify patterns of conserved genes. Using OrthoFinder, a total of 9,715 homologous gene families were identified in NREL 46B-D3, including 2,883 gene families conserved across all the Chlorophyta genomes included in this analysis. Gene family expansion and contraction analysis using CAFE showed that 653 gene families are expanded and 1,216 gene families are lost in the NREL 46B-D3 genome. Three-thousand six-hundred and sixty-nine single-copy genes and 275 multi-copy gene families were unique to NREL 46B-D3 (Supplementary Data File 3). While the majority of these unique gene families lack any annotation, some species-specific or genus-specific families contain protein domains related to DNA-binding and transcriptional regulation, including bZIP domains and MYND Zinc fingers. Other unique gene clusters include a couple of ABC transporters, protein kinases, and several enzymes (citrate synthase, chalcone synthase, triglyceride lipase, NADH dehydrogenase I).

Presence of Pfam domains were investigated in the NREL 46B-D3 genome and compared to other Chlorophyta genomes. There

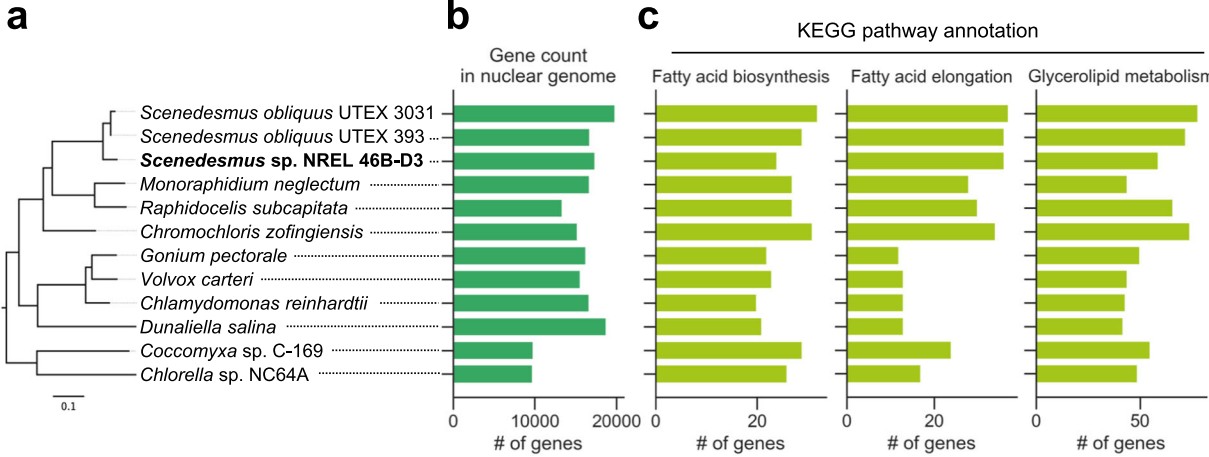

**Fig. 3 Genome comparison of *Scenedesmus* sp. NREL 46B-D3 and related algae. a** Maximum-likelihood phylogenetic tree of *Scenedesmus* sp. NREL 46B-D3 and related algae based on 606 single-copy orthologous genes. Bootstrap support values on all branches are 100%. Scale bar shows the mean number of nucleotide substitutions per site. **b** Number of predicted genes models in the nuclear genomes of related algae. **c** Number of genes assigned to lipid-related KEGG pathways: fatty acid biosynthesis, fatty acid elongation, and glycerolipid metabolism.

were 3,200 distinct Pfam domains identified in the NREL 46B-D3 genome (7,304 out of 17,399 genes were annotated with at least one Pfam domain), which is slightly fewer to the number found in the *S. obliquus* UTEX 3031 genome (3,363 distinct Pfam domains) and in UTEX 393 (3,520). Five Pfam domains were only found in the NREL 46B-D3 genome and missing in all other Chlorophyceae genomes analyzed. These Pfam included domains of unknown function (DUF616 and DUF4573) and a BAAT/Acyl-CoA thioester hydrolase C-terminal domain (PF08840). There were an additional 79 Pfam domains specific to the genus *Scenedesmus* (i.e., present in UTEX 3031, UTEX 393 and NREL 46B-D3 but not in other algae used in this analysis), including viral genes and genes of bacterial origin (Supplementary Data File 3).

Transporter genes that were functionally assigned based on the Transporter Classification Database (TCDB)[30] were compared across genomes. A total of 940 genes in the NREL 46B-D3 genome were identified as transporters by BlastP against TCDB, and the number of gene copies of individual transporters varied across *Scenedesmus* genomes. There were two copies of the endosomal K+ /H+ antiporter gene (TCDB 2.A.37.4.2) present in the NREL 46B-D3 genome (Protein IDs: 1565674 and 1636933) and one copy in the other two *Scenedesmus* genomes. Similarly, there were three copies of the ClC-1 voltage-gated Cl channel gene (TCDB 2.A.49.6.1) in the NREL 46B-D3 (Protein IDs: 1493535, 1521651, 1549247) and one copy in the other Chlorophyta genomes. The gene annotated as a high-affinity K+ uptake transporter (TCDB 2.A.72.3.3) was present in the NREL 46B-D3 genome (Protein ID: 1699663), but not in the other *Scenedesmus* genomes.

Based on Pfam domains and PlantTFDB families[31] a total of 215 transcription factors (TF) in 161 gene families have been predicted in the NREL 46B-D3 genome. The most common TF families present in the genome are the MYB-like, SBP, bZIP, and C3H families. The largest TF gene family present consists of 23 TF genes with a MYB-like DNA-binding domain and while conserved across the other Chlorophyta genomes, it is expanded in Sphaeropleales genomes. Forty-eight TF gene families are conserved across Chlorophyta genomes. There are 31 TF families present only in the *Scenedesmus* genomes, and of these, 11 TF gene families are unique to the NREL 46B-D3 strain (Supplementary Table 2). TF expression changes were observed in both cold and heat stress sample profiles, reflecting an aspect of the global changes driven by temperature stress. Thirty-eight percent

of all the TFs (83) had decreased expression in the cold stress while 22% (47) had decreased expression in the heat stress. Thirty-five TFs were downregulated in both conditions. Fifteen percent of TFs (33) had increased expression in the cold stress and 10% (21) had increased expression in the heat stress. Only five TFs were upregulated in both conditions.

To explore differences in metabolism revealed by gene content, we compared the number of enzymes present across metabolic pathways based on EC numbers predicted by Priam[32] (Supplementary Table 3). While the total number of unique EC numbers are roughly comparable across Sphaeropleales and Chlamydomonales genomes, the number of predicted genes increased within certain KEGG pathways (Fig. 3c and Supplementary Data File 4). In particular, the number of fatty acid elongation pathway genes increased in Sphaeropleales genomes compared to Chlamydomonales genomes. The number of fatty acid elongation pathway genes in *Scenedesmus* sp. NREL 46B-D3 is twice that of *C. reinhardtii*, while the number of genes in the KEGG pathways for fatty acid biosynthesis and glycerolipid metabolism are slightly higher for *Scenedesmus* compared to *C. reinhardtii* and other Chlamydomonales genomes. Furthermore, genes involved in the carotenoid biosynthesis pathway were identified in the NREL 46B-D3 genome (Supplementary Table 4), and the pathway appears to be similar to those of other Chlorophytes, including *Chlamydomonas reinhardtii*.

**Metabolite levels in control and temperature stress conditions.** Growth of the NREL 46B-D3 strain was monitored by optical density prior to and throughout temperature stress (Fig. 4 and Supplementary Data File 5). During the 24-h sampling period and the 4 days following cold stress exposure, the cold stress cultures showed slower growth compared to the control and heat stress cultures.

Over the 24-h sampling period, the metabolite levels of lipids and polar metabolites were measured by liquid chromatography tandem mass spectrometry (LC/MS-MS) at 4-h intervals in the control and temperature stress samples. Lipidomics profiling of triacylglycerol (TAGs), monogalactosyldiacylglycerols (MGDGs), and digalactosyldiacylglycerols (DGDGs) detected a broad range of molecular species (Fig. 5a–c and Supplementary Data File 6). While TAG levels broadly increased across the first five time points of the cold stress samples and increased in the last two

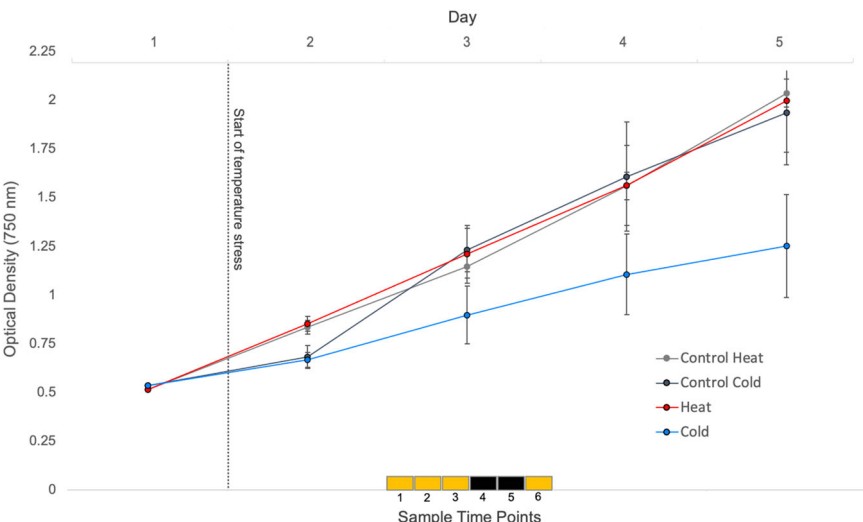

**Fig. 4 Growth according to optical density measurements of *Scenedesmus* sp. NREL 46B-D3.** The dotted line indicates the start of temperature stress. The upper x-axis displays days while the squares on the lower x-axis indicate the six sample time points and the respective photoperiod (yellow = light cycle, black = dark cycle). Data points represent the average and error bars show the standard deviation of $n = 3$ biological replicates.

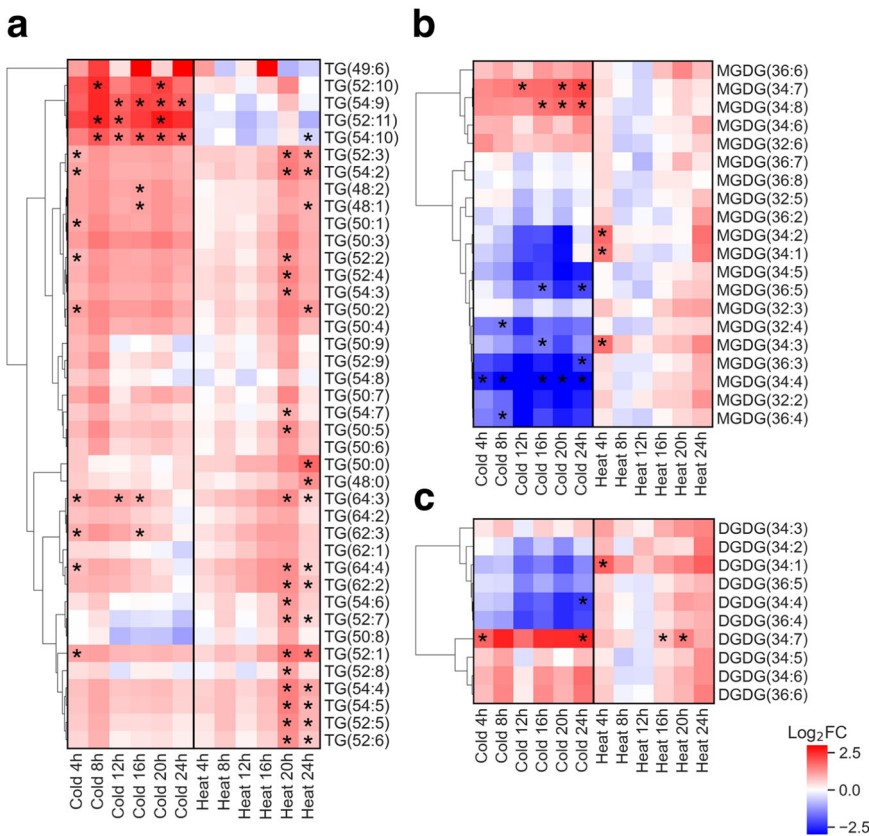

**Fig. 5 Fold change of lipid levels in cold stress and heat stress samples.** Heatmaps show fold change over time in **a** triacylglycerol (TG) levels, **b** monogalactosyldiacylglycerol (MGDG) levels, and **c** digalactosyldiacylglycerol (DGDG) levels of the highest abundance species in cold stress and heat stress samples relative to the control samples at the corresponding time point. Labels for molecular species indicate the number of carbons in the fatty acyl chains followed by the number of double bonds. Asterisks (*) indicates p-value < 0.05. Rows are ordered by hierarchical clustering of mean fold change over three replicates.

time points of the heat stress samples, there were several polyunsaturated TAGs (52:10, 52:11, and 54:10) that were higher only in the cold stress samples. Of the high abundance MGDG species, MGDG (34:7) levels increased in the cold stress, whereas levels of those MGDG species with fewer double bonds, MGDG

(34:2, 34:3, 34:4, 34:5), decreased in the cold stress. Similarly, for DGDG species, DGDG (34:7) levels increased in cold stress and DGDG (34:4) decreased in the cold stress. In general, the lipids with a higher degree of desaturation increased in abundance in the cold stress samples relative to the control.

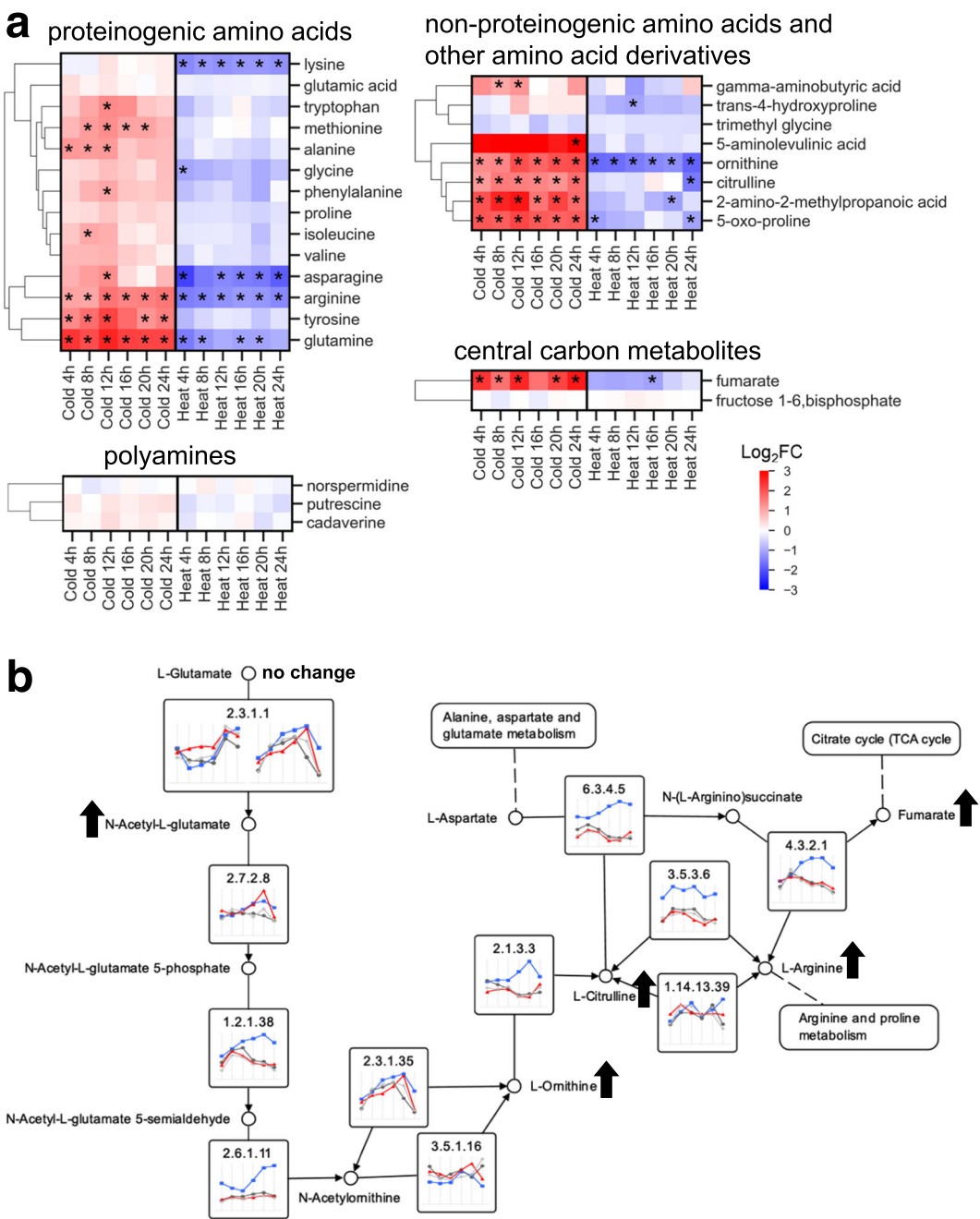

**Fig. 6 Amino acid metabolism under cold stress and heat stress. a** Heatmap of fold change in metabolite levels over time in cold stress and heat stress samples relative to the control samples at the corresponding time point. Asterisk (*) indicates *p*-value < 0.05 and absolute Log₂FC > 1. Rows are ordered by hierarchical clustering of mean fold change over three replicates. **b** Pathway map based on KEGG EC annotations for the arginine biosynthesis pathway. Gene expression profiles (FPKM over time) are shown by gene in cold stress (blue), heat stress (red), and controls (dark gray and light gray) averaged over three replicates each. Arrows indicate the change or no change in the cold stress sample relative to the control sample for the detected metabolites.

Elevated amino acid levels were observed in the cold treatment compared to the control, while the heat treatment showed decreased amino acid accumulation (Fig. 6a and Supplementary Data File 7). In particular, ʟ-asparagine, ʟ-glutamine, ʟ-lysine, ʟ-ornithine, and ʟ-arginine decreased in the heat stress samples, whereas the cold stress samples showed an increase in amino acids across different classes, including arginine, aromatic amino acids, methionine, alanine, glutamine, and asparagine. Metabolites involved in arginine biosynthesis, such as ornithine, citrulline, and N-acetyl-ʟ-glutamate also showed an increase in abundance in the cold stress samples (Fig. 6b and Supplementary Data File 7). However, only a slight

increase (Log₂FC < 1) was observed in proline, glycine, and valine. Polyamines showed minimal changes in abundance in response to temperature stress. Other amino acid derivatives, such as gamma-aminobutyric acid (GABA), 5-oxo-proline, 2-amino-2-methylpropanoic acid, also significantly increased in the cold stress samples.

Two central carbon metabolism metabolites were detected by LC/MS-MS. Fumarate, a TCA cycle intermediate, significantly increased in the cold stress samples and decreased slightly in the heat stress samples relative to the controls (Fig. 6a and Supplementary Data File 7). Fructose 1,6-bisphosphate showed no changes over the sampling period in either condition.

In addition, untargeted metabolomics analysis was used to identify features that matched 177 unique compounds in the Global Natural Products Social Molecular Networking spectral library[33]. Amongst these library matches, features were detected in the control and temperature stress samples that matched similar MS/MS spectra of carotenoid pigment molecules, astaxanthin, beta-cryptoxanthin, and canthaxanthin (Supplementary Fig. 5). Evidence of carotenoids in the metabolomics samples is consistent with the pigmentation observed in the micrograph (Fig. 1a).

**Transcriptomic response**. Next, we investigated the global transcriptional response over the 24-h period following temperature stress (Fig. 7 and Supplementary Data File 8). Differential expression analysis was performed using DESeq2 comparing the treatment samples to the control samples at the corresponding time points. A gene was considered differentially expressed between conditions when the adjusted $p$-value < 0.05 and $|Log_2FC| > 1$. Cold stress samples demonstrated a higher number of differentially expressed genes, with 3,976 genes increased in expression (about 22% of total genes) and 5,284 genes decreased in expression (about 30% of total genes) relative to the control. By contrast, in the heat stress samples, 2,168 genes increased in expression (about 12% of total genes) and 3,460 genes decreased in expression (about 20% of total genes). Principal component analysis (PCA) of the transcriptomics data showed that the cold stress samples were distinct from the heat and controls. The cold stress samples from all time points clustered closely while heat stress samples did not show a distinct transcriptomics profile from the control samples (Fig. 7a and Supplementary Data File 9).

To analyze the global patterns of expression, we constructed a co-expression network produced by the pairwise correlations between genes computed by Weighted Gene Co-expression Analysis (WGCNA)[34], revealing two major groups of genes in the network structure (Fig. 7c and Supplementary Data File 10). The first group, which are the genes on the left side of the co-expression network, consists of genes that decreased in expression in the cold stress samples compared to control samples. The second group, which is the larger group of genes on the right side of the network, is mostly composed of genes that increased in expression in the cold stress samples, including a subset of genes that had lower expression in both stress conditions compared to control samples. WGCNA identified 33 gene modules with correlated co-expression profiles calculated across time points and conditions (Supplementary Data File 11). The gene modules on average contain 218 genes and with a range in size from 52 genes to 954 genes. Functional enrichment by GO terms was calculated for each gene module. Several of the co-expressed gene modules were functionally enriched with GO terms associated with lipid metabolism. There were two gene modules that were enriched for genes assigned the GO term for fatty acid biosynthetic process (GO:0006633). One of these gene modules contains lipid metabolism-related genes, such as enoyl-ACP reductase (*ENR1*, Protein ID: 1470501), ACP S-malonyltransferase (*MCT1*, Protein ID: 1481491), acetyl-CoA biotin carboxyl carrier (*BCC1*, Protein ID: 1520010), beta-carboxyltransferase (*BCX1*, Protein ID: 1701104), and three fatty acid desaturase genes, *FAB2*, *FAD7*, and *FAD2* (Protein IDs: 1284617, 1285947 and 1476226). In addition, the module is enriched with genes involved in photosynthesis (GO:0015979), glycolysis (GO:0006096) and gluconeogenesis (GO:0006094) (module is colored tan in Fig. 7c). Several enzymes involved in starch metabolism are also members of this module and increased slightly in expression in cold stress samples

(Supplementary Fig. 6). The other gene module is enriched with genes involved in translation (GO:0006412), cell redox homeostasis (GO:0045454), and some amino acid metabolic processes, such as glycine metabolic process (GO:0006544), aromatic amino acid family biosynthetic process (GO:0009073), methionine biosynthetic process (GO:0009086), and serine metabolic process (GO:0006563) (module is colored blue in Fig. 7c). This module contains lipid metabolism genes like alpha-carboxyltransferase (*ACX1*, Protein ID: 1281933), acyl carrier protein (*ACP1*, Protein ID: 1268970), beta-ketoacyl-ACP synthase III (*KAS3*, Protein ID: 1496856), 3-hydroxyacyl-ACP dehydratase (*HAD1*, Protein ID: 1539120), oleoyl-ACP thioesterase (*FAT1*, Protein ID: 1561707), 3-oxoacyl-ACP reductase (*KAR1*, Protein ID: 1622466), and Lecithin:cholesterol/phospholipid:diacylglycerol acyltransferase (*LCAT*, Protein ID: 1139915). For both of these modules, the general expression pattern shows increased expression in the cold stress samples, suggesting that genes in these modules are cold-responsive.

Up- and downregulation of genes varied by metabolic pathway (Supplementary Fig. 7). In the cold stress samples, the largest changes were observed in amino acid metabolism genes compared to the other metabolic pathway categories. For example, after 24 h, there were 71 upregulated genes in amino acid metabolism, compared to only 33 and 24 genes upregulated in nucleotide and carbohydrate metabolism, respectively. Similar to the overall response, fewer genes were upregulated or downregulated in the heat stress samples across the different metabolic pathway categories. In contrast to the cold stress response, fewer than ten genes were upregulated in amino acid metabolism in the heat stress sample. The number of genes downregulated by pathway were similar between the heat stress and cold stress samples.

Many TFs exhibited cold-specific expression changes. Twenty-eight upregulated and 48 downregulated TFs were differentially expressed exclusively in cold stress samples (Fig. 7b). A co-expression module enriched with glycolysis, photosynthesis, fatty acid metabolism genes, and a common MYB DNA-binding TF (Protein ID: 1747342), part of the genome's most ubiquitous MYB-like TF family, was identified. This TF showed increasing differential expression for the first four time points and averaged 1.2 $Log_2FC$ across all six-time points relative to the control. An additional co-expression module, also enriched with fatty acid metabolism genes, identified a NY-FB subunit, a B3-type, and two C2H2-type zinc finger TFs (Protein IDs: 1286905, 1699981, 1578187, and 1751936, respectively) that showed slightly increased expression for the cold samples, averaging 0.5 to 1.0 $Log_2FC$ (Supplementary Fig. 8).

We also identified the most down- and upregulated TFs independent of the co-expression modules. The most upregulated TF (Protein ID: 1524442), which contains a high mobility group (HMG) box domain, was overexpressed following the first time point (4 h) with an average 1.3 $Log_2FC$ in the cold samples relative to the control. Often occurring in combination with the HMG-box domains of plants, an ARID TF (Protein ID: 1101024) was also upregulated, potentially in tandem with HMG-box ($R^2 = 0.81$). Conversely, the most downregulated TF (Protein ID: 1267608) in the cold stress samples contained a Zinc finger C-x8-C-x5-C-x3-H type (Znf CCCH-type) domain belonging to the C3H TF family. It was consistently expressed at low levels across all time points with an average $-1.9$ $Log_2FC$ as well as a MYB DNA-binding TF (Protein ID: 1768631) with an average $-1.8$ $Log_2FC$ relative to the control.

**Changes in expression of genes involved in fatty acid metabolism**. Several key genes involved in fatty acid biosynthesis were

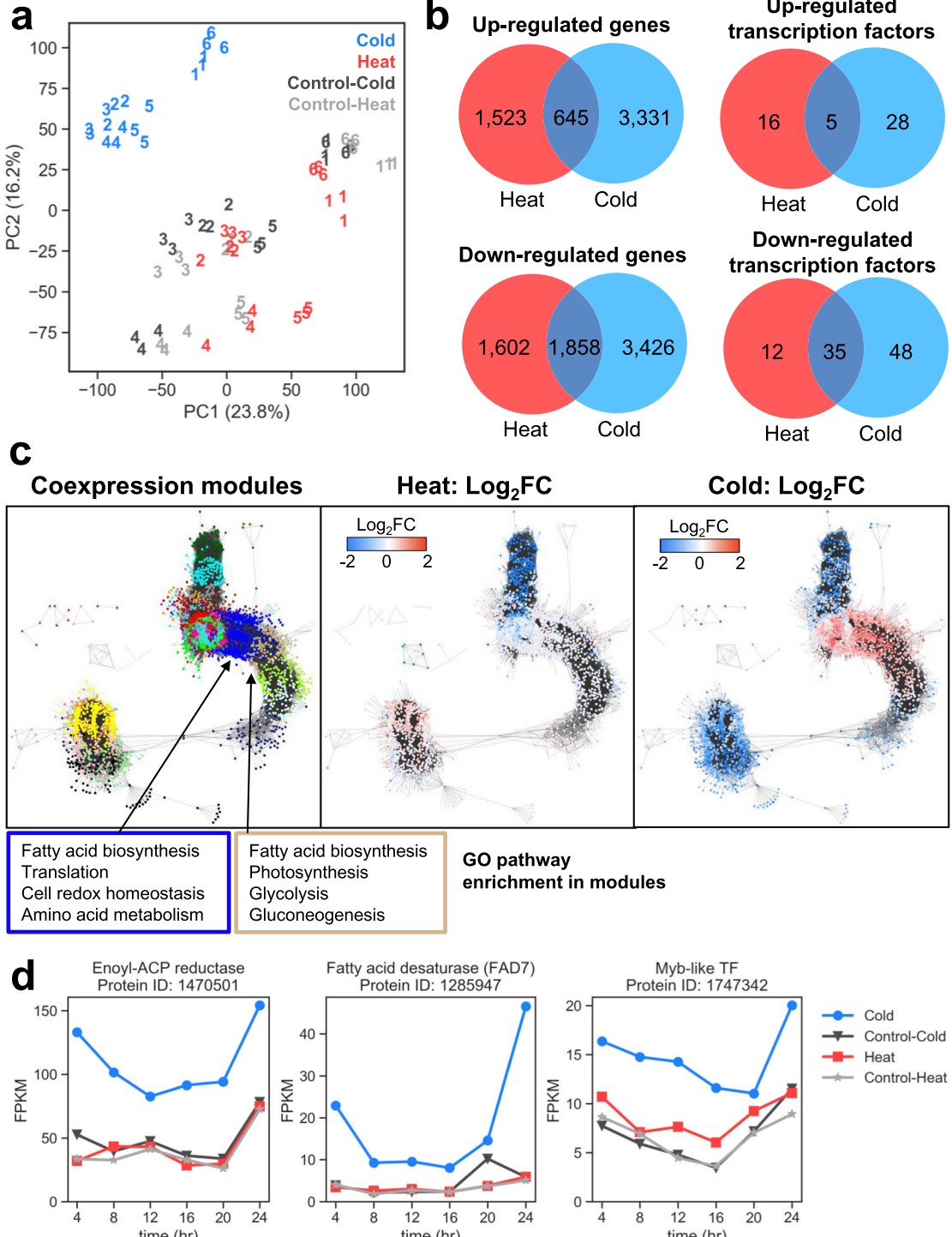

**Fig. 7 Transcriptomics profile of *Scenedesmus* sp. NREL 46B-D3 in response to cold stress and heat stress conditions. a** A principal component analysis of the gene expression data colored by condition: cold stress (blue), heat stress (red), and controls (gray) and labeled by time point, with three replicates for each. The time points correspond to the following times over the sampling period: 1–4 h, 2–8 h, 3–12 h, 4–18 h, 5–20 h, 6–24 h. **b** Venn diagram of differentially expressed transcription factors in at least one time point across the cold stress and heat stress samples compared to the control samples. **c** Gene co-expression network calculated using weighted gene co-expression network analysis (WGCNA) with genes colored by modules (left), average fold change in the heat samples (middle), and average fold change in cold samples (right). Enriched GO processes for modules that are enriched with fatty acid biosynthesis genes are listed, and the borders are colored by modules shown in the co-expression network. **d** Gene expression profiles over time for several upregulated genes in cold stress, including enoyl-ACP reductase (*ENR1*), omega-3 fatty acid desaturase (*FAD7*), and a MYB-like transcription factor.

consistently upregulated in cold stress samples (Supplementary Fig. 9a). The genes that encode the individual components of the fatty acid synthase complex, 3-ketoacyl-CoA synthase (*KAS1*, Protein ID: 1523986), 3-ketoacyl-ACP synthase (*KAS2*, Protein ID: 1293235), and β-ketoacyl synthase (*KAS3*, Protein ID: 1496856), showed increased expression in the cold stress samples. Another gene involved in fatty acid metabolism that showed an elevated response in cold stress was the enoyl-ACP reductase gene (*ENR1*, Protein ID: 1470501), encoding for FabI, the fatty acid synthase that catalyzes fatty acid elongation through the reduction of enoyl-ACP to the saturated acyl-ACP, leading to TAG formation. The upregulation of *ENR1* was previously observed in *Neochloris oleoabundans* during salinity stress, which resulted in increased lipid concentration[35]. Changes in the expression of genes involved in fatty acid synthesis were only observed in the cold stress but not the heat stress, suggesting that fatty acid synthesis is only modulated in *Scenedesmus* sp. NREL 46B-D3 in response to cold under the conditions tested here.

In addition to changes in the expression of genes involved in fatty acid synthesis, changes in the expression of fatty acid desaturation enzymes were also observed in both cold and heat stress samples. Fatty acid desaturases (FADs) catalyze the desaturation of fatty acids that result in the addition of a carbon-carbon double bond at specific positions in the acyl chain. There were 15 annotated FAD genes in the *Scenedesmus* sp. NREL 46B-D3 genome, and several exhibited changes in expression in response to both cold and heat stress (Supplementary Fig. 9b). One FAD gene (*FAD5*, Protein ID: 1505577), a putative ortholog of the MGDG-specific palmitate delta-7 desaturase[36], consistently increased in expression in the cold stress and increased in expression in the heat stress at middle time points, but was not co-expressed with genes of similar function.

Six FAD genes showed a cold-specific response, and increased expression of these FAD genes is correlated with increased abundance of MGDGs and DGDGs with a lower degree of saturation. In particular, the omega-3 fatty acid desaturase (*FAD7*, Protein ID: 1285947) gene was significantly upregulated in the cold stress with an average of 1.9 Log$_2$FC in cold stress samples. The *FAD7* gene was co-expressed within the same module as several other lipid-related genes that exhibit a cold-specific response, including *ENR1*, *KAS2*, *KAR1*, and *ACX1* (Protein IDs: 1470501, 1293235, 1622466, and 1281933). In this co-expression module, a stearoyl-ACP 9-desaturase (*FAB2*, Protein ID: 1284617) had an average Log$_2$FC of 1.2 in cold stress, and a delta-12 desaturase (*FAD2*, Protein ID: 1476226) had an average Log$_2$FC of 0.8 in cold stress. Additionally, these FAD genes were co-expressed with the aforementioned MYB-like TF (Protein ID: 1747342) and may be involved in their regulation.

**Changes in expression of genes involved in amino acid metabolism.** Co-expression gene modules that showed increased expression patterns in the cold stress were enriched for genes involved in amino acid biosynthesis pathways. Genes involved in glutamate and aspartate synthesis, precursors for other amino acids, increased in expression in cold but not heat stress samples (Supplementary Fig. 10a). Glutamine synthetase, a key enzyme involved in nitrogen assimilation, catalyzes the formation of glutamine through the condensation of glutamate and ammonia. The glutamine synthetase gene (EC 6.3.1.2, Protein ID: 1500976) was upregulated in the cold stress samples with a mean Log$_2$FC of 1.0 and did not show a significant change in expression in the heat stress samples.

The gene encoding cobalamin-independent methionine synthase (EC 2.1.1.14, Protein ID: 1182669) dramatically increased in expression under cold stress across all time points. Methionine synthase is responsible for transferring a methyl group to L-homocysteine to produce L-methionine, which can be converted to S-adenosyl-L-methionine. Argininosuccinate synthase (EC 6.3.4.5, Protein ID: 1552779), which catalyzes a step in the arginine biosynthesis pathway to form argininosuccinate from citrulline and aspartate, increased in expression in the cold stress (Fig. 6b and Supplementary Data File 12). In the polyamine biosynthesis pathway, genes encoding ornithine decarboxylase (EC 4.1.1.17, Protein ID: 1293672 and 1293843) and agmatine deiminase (EC 3.5.3.12, Protein ID: 1704069) significantly decreased in expression in the cold stress samples. In addition, genes encoding key enzymes involved in proline synthesis, such as both copies of the pyrroline-5-carboxylate reductase gene (EC 1.5.1.2, Protein ID: 1279716 and 1694464), were upregulated up to Log$_2$FC of 1.8 after 24 h.

Along with increased expression of amino acid metabolism genes, aminoacyl-tRNA ligase genes are upregulated in the cold stress samples, many of which belong to the co-expression module enriched with genes involved in translation. The increased expression of aminoacyl-tRNA ligase genes is broad and highest at the 4 and 24-h time points (Supplementary Fig. 10b). Only two of the twenty types of aminoacyl-tRNA ligase genes (His and Phe) did not significantly increase in expression at any time points.

## Discussion

Strain screening is an effective strategy for identifying candidate microalgae with favorable characteristics for biofuel feedstocks. Of particular interest are specific "seasonal" strains that can be rotated in order to maximize biomass production throughout the year. Screening under simulated Summer temperature cycling indicated the highest biomass accumulation potential for *Scenedesmus* sp. NREL 46B-D3 relative to the other 106 strains tested. Notably, this strain also emerged as a top-candidate strain in prior simulated Winter temperature strain screening efforts, indicating robust productivity across broad temperature regimes. Follow-on studies herein revealed substantial biomass accumulation in the days following cell division. Indicative of photosynthetic activity in a nondividing state, this enables carbon sequestration and storage carbon accumulation in the absence of cell division. Thus, we hypothesize that the high biomass accumulation phenotype, relative to other strains evaluated during screening, is due in part to this high storage carbon accumulation phenotype post cell division. This phenotype of concurrent lipid and carbohydrate accumulation post cell division has development potential for a genetically engineered photosynthetic biocatalyst, wherein sustainable multi-day production can be achieved without nutrient input requirements.

Maintaining a productivity of >16 g/m$^2$/day post-cell division, NREL 46B-D3 accumulates storage carbon almost exclusively in the form of lipids and carbohydrates that can be readily converted into an array of fuel and chemical intermediates. A similar phenotype has been previously observed in other algae, including a top-candidate deployment freshwater strain, *Scenedesmus obliquus* UTEX 393[23,24,37], which prompted us to do a direct comparison to NREL 46B-D3 (Fig. 1c). In this comparison, *Scenedesmus* sp. NREL 46B-D3 accumulated over 2x the biomass and storage carbon when cultivated in media representing half-strength seawater (at 17.5 g/L media salinity). Although *Scenedesmus obliquus* UTEX 393 is generally considered a freshwater strain, this underscores *Scenedesmus* sp. NREL 46B-D3 halotolerance and sustainability potential by reducing demands on freshwater resources. Further, this strain's biomass titers observed at lower salinities (0.5X seawater) were even higher than those

observed at full strength seawater (Supplementary Fig. 1b), indicating potential for brackish water utilization, obtaining higher productivities in regions with abundant low-salinity water sources. Collectively, this strain's characteristics show high potential for genetic enhancement and deployment in outdoor cultivation.

The *Scenedesmus* sp. NREL 46B-D3 genome and transcriptome provide further insight into the metabolic and transcriptional stress response. While our current understanding of the molecular pathways in *Scenedesmus* species is limited, it is growing due to the genomic sequencing of several *Scenedesmus* species with biofuel feedstock potential[24,28,38] and additional transcriptomics profiling experiments[29,39]. Differences in the number of ion transporter gene copies in the NREL 46B-D3 genome compared to the *S. obliquus* UTEX 393 and *S. obliquus* UTEX 3031 genomes may be related to its halotolerance, including the endosomal K +/H+ antiporter and a voltage-channel chloride channel. Members of the K+/H+ antiporter family have been shown to be induced by salt stress in plants[40]. The high-affinity K+ uptake transporter gene (Protein ID: 1699663), missing in other *Scenedesmus* genomes, belongs to the HAK/KUP/KT transporter family, including members that contribute to salt tolerance in plants[41].

In addition to transporter gene content differences, the NREL 46B-D3 genome exhibits differences in TF gene families. Our comparative analysis found multiple TF gene families present only in the NREL 46B-D3 genome. Increasing TAG content through targeted modification of TFs involved in lipid synthesis has shown to be a successful strategy, but algal TF identification and functional characterization is largely unexplored[42]. However, phylogenetic relatedness can be indicative of TF-family profile[35], and broadly conserved TFs, particularly those that are stress-responsive and regulate carbon partitioning, may be critical targets for strain engineering across algal taxa. In the NREL 46B-D3 genome, MYB-like TFs were the largest TF gene family with 36 MYB-like genes, a number comparable to the 35 MYB-like TF genes in the *Nannochloropsis oceanica* IMET1 genome[35]. In plants, MYB-like TFs are involved in key mechanisms including stress responses to cold, phosphorus and nitrogen starvation and UV radiation[43,44]. While most have not been functionally characterized, MYB-like TFs may play a similar role in algae. Leveraging the co-expression modules, a MYB-like TF (Protein ID: 1747342) responsive to cold stress was indicated in glycolysis, photosynthesis, and fatty acid metabolism regulation. Additionally, a co-expression module of NREL 46B-D3 enriched with fatty acid metabolism genes showed increased C2H2-type zinc finger TF (Protein ID: 1751936) expression during cold stress in concert with a NF-YB subunit (Protein ID: 1286905), whose family members are known to enhance plant tolerance to drought, salinity and cold stress[45]. Also identified was a B3 type TF (Protein ID: 1699981), whose family members synchronize *Arabidopsis* growth with seasonal temperature changes[46]. Targeted modulation of NREL 46B-D3 TFs related to stress response and fatty acid synthesis could be advantageous. However, overexpression without TF characterization, including the identification of the regulated gene and the metabolic outcome, could prove detrimental, as previously observed in *Arabidopsis*[47]. These observations, coupled with algal TF domains that do not occur in plants[48], emphasize the need to characterize algal TFs when identifying targets for strain improvement.

Microalgae lipid composition, including the proportion of saturated and polyunsaturated fatty acids, is often influenced by temperature[49,50]. Changes in fatty acid desaturase gene expression could affect the fraction of polyunsaturated fatty acids. We have previously observed alterations to *Scenedesmus* sp. NREL 46B-D3 degree of fatty acid saturation under cold cultivation conditions[4]. Our functional genomic analyses provide insight into the putative mechanism and certain FADs driving fatty acid saturation under cold stress. Specifically, the omega-3 fatty acid desaturase FAD7 gene increased expression in the cold stress, consistent with increased MGDGs and DGDGs lipids with polyunsaturated fatty acids. The cold-responsive behavior of FAD7 has been observed across other green microalgae[51–54]. Shifts in thylakoid membrane lipid composition may be an important mechanism for cold tolerance. In addition, controlling the degree of lipid saturation could facilitate optimization of biofuel composition.

Amino acids role and response to abiotic stress varies across algal strains. Elevated amino acid concentrations in response to stress have been observed in plants[55] and algae[12,15,42,56] and was observed in NREL 46B-D3 in response to both heat and particularly the cold stress. General amino acid accumulation is a mechanism to relieve oxidative stress, which can be caused by hypothermia and the resulting the generation of $H_2O_2$ in plants as well as *Chlamydomonas*[57–59]. Specifically, under stress the cell preemptively reserves nitrogen in TCA intermediates for subsequent synthesis of particular amino acids, including arginine[60]. Arginine minimizes the impact of oxidative stress by amplifying an overall stress response and is key for accessing nitrogen stores during nitrogen deprivation while serving as a precursor for proline[60]. We observed significantly higher concentrations of arginine and increased expression of several arginine biosynthesis genes in the cold stress. Similarly, *Chlamydomonas reinhardtii* showed a transient accumulation of particular amino acids during cold stress, including arginine over a 48-h period[61]. Proline accumulation is a common stress tolerance mechanism in algae and plants[62,63]. We observed only a slight proline increase and an increase in proline biosynthesis gene expression in the cold stress samples. Significant increases of glutamine, orthinine, citrulline, and tyrosine were also observed in all six cold stress samples. These observed increases in various types of amino acids suggest amino acid production may play an important role in NREL 46B-D3 cold stress response.

Directly synthesized from three amino acids including arginine, polyamines are often implicated in stress tolerance in algae and plants[55,64–66], we did not observe significant changes in spermidine, putrescine, or cadaverine levels in cold or heat stress samples. However, several enzymes involved in polyamine synthesis, particularly in the cold stress samples were modulated, such as decreased expression of ornithine decarboxylase encoding genes, the rate-limiting step in polyamine synthesis[67]. In addition to polyamines, gamma aminobutyric acid (GABA) accumulation is a common plant stress response[68]. GABA accumulation has not been broadly studied in algae, but increased GABA in response to desiccation stress was observed in *Trebouxia* species[69]. In *Scenedesmus* sp. NREL 46B-D3, the increase in GABA was significant at several time points in the cold stress relative to the control but less than other metabolites. Further studies are needed to determine if polyamines and GABA contribute to a physiological adaptation to cold stress in *Scenedesmus* sp. NREL 46B-D3.

## Conclusion

Microalgae carry a great potential for the production of fuels and chemicals challenged by various economic and sustainability issues. To overcome these issues, we developed a pipeline for multi-factorial characterization of new algal strains, including growth profiling, genome sequencing, and multi-omics measurements translated into predictive models to identify gene targets for strain improvement.

Here, we have characterized the growth, temperature, salinity tolerance, and composition of *Scenedesmus* sp. NREL 46B-D3.

Peak growth at 25 °C indicates this strain is ideal for temperate region cultivation and its maximal productivity (30 g/m²/day) exceeds that reported by Davis et al.[70] for cost-competitive microalgal biofuels. Therefore, this green alga has substantial biofuel feedstock potential with robust traits required for outdoor deployment, including high biomass accumulation capacity, stationary phase photosynthetic and storage carbon accumulation capacity, and a high lipid and carbohydrate percentage (80% AFDW).

To better understand its biology and enable strain improvement, genomic, transcriptomic, and metabolomic characterization under cold and heat stress were used to identify temperature-responsive genes, including several TFs. Differences in the presence of transporter genes across *Scenedesmus* genomes may explain differences in strain-specific phenotypes like halotolerance. Furthermore, changes in the lipid composition and amino acid levels were observed in response to cold stress, and these metabolic changes coincided with increased expression of genes encoding specific metabolic enzymes such as fatty acid desaturases. Understanding changes on the gene level and metabolism in response to temperature perturbations in algae will help us engineer more robust and productive strains. Future development of molecular tools for *Scenedesmus* sp. NREL 46B-D3 will facilitate specific gene target modulation and ultimately strain improvement.

## Methods

**Strain selection, characterization, and identification**. One-hundred seven halotolerant algal strains were screened under simulated summer conditions using the same methods as reported by Dahlin et al.[3]. Briefly, 100 mL cultures were sparged with 100 mL/min with air containing 2% CO₂, temperature cycles from 21 to 32 °C, while lighting cycles from 0 to 965 μmol/m²/s, as described previously in Arora et al.[71]. These were designed to simulate the observed summer (June 12th to July 21st, 2014) outdoor conditions of 1,000 L algal ponds at the Arizona Center for Algae Technology and Innovation testbed site located in Mesa, Arizona. Screening utilized a modified f/2 medium, termed NREL Minimal Medium (NM2) as described previously[3], with the exception of data presented in Figs. 2 and 1c in which temperature was a constant 25 °C, with a 12:12 light:dark cycle, 350 μmol/m²/s light intensity, and replacement of media ammonium chloride with ammonium bicarbonate. The experiment relating to Fig. 2 utilized a salinity of 8.75 g/L sea salts, while the experiment relating to Fig. 1c utilized a salinity of 17.5 g/L sea salts. Additional modifications to temperature, light intensity, and media components were implemented for subsequent characterization detailed in Supplementary Fig. 1.

Temperature optimum data utilized the above screening methods, except temperature and light (400 μmol/m²/s) were held constant. Salinity tolerance data were generated similar to the above screening methods, except the basal NM2 media was generated with half seawater and half Milli-Q water (Millipore Corporation), to generate the 17.5 g/L salinity media. Higher salinities were obtained by adding sea salts (Sigma S9883) to the appropriate final salinity concentration. Varying nitrogen to phosphorus ratios utilized the above screening methods, except sodium nitrate was utilized as a nitrogen source instead of ammonium chloride. Nitrogen levels were held constant at 5 mM, while phosphate concentrations were decreased to achieve the appropriate ratio reported. Compositional analysis (FAME, carbohydrate, ash) was conducted as reported by Dahlin et al.[4]. Data relating to Figs. 1c and 2, Supplemental Fig. 1b, c, and Supplemental Fig. 2 utilized two biological replicates, as the range of variables tested and time points sampled were prioritized over additional replicates, reflecting previous methodologies[3,72].

**Temperature challenge studies**. In order to evaluate the effect of suboptimal outdoor temperature, *Scenedesmus* sp. NREL 46B-D3 was grown in summer temperatures (May, June, and July, 2014) that were observed during outdoor cultivation in Mesa, Arizona at the Arizona Center for Algae Technology and Innovation testbed site. The average afternoon temperature, 28.2 °C, was selected as the control temperature and within the determined optimal temperature range (Supplementary Fig. 1). Subsequently, the heat stress, 33.2 °C, was the observed high temperature while the cold stress, 13 °C, was the observed low from the same time period. Two-liter Erlenmeyer flasks were inoculated with f/2 medium[73] and culture at the same optical density (0.5 at 750 nm) for a total volume of 650 mL. Flasks were placed on shaker tables (125 rpm) in growth chambers, with a 18:6 light:dark cycle (300 μmol/m²/s), maintaining a 3,000 ppm CO₂ concentration, a constant control temperature setting of 28.2 °C for 24 h. After 24 h, triplicates were cultivated in growth chambers under identical conditions with the exception of temperature, which was set at either control (28.2 °C), cold (13 °C), or heat (33.2 °

C) settings for an additional 24-h adjustment period. Following this adjustment period, samples for transcriptomic and metabolomic analysis were collected every 4 h for 24 h. Following centrifugation, 20–50 mg of biomass was frozen in liquid-nitrogen and stored at −80 °C for subsequent transcriptomic and metabolomic analysis. Optical density, verified with cell counts using a hemocytometer, was used to determine growth for a total of 5 days.

**DNA extraction**. Genomic DNA (gDNA) was extracted using gentle digestion and purification in agarose plugs. In brief, algal cells were washed, resuspended in the buffer and combined with an equal volume of 1% low-melting point agarose. The resulting mix was pipetted into plug molds. The solidified plugs were incubated overnight in a protoplasting solution containing hemicellulase and driselase enzymes (final concentration 40 and 20 mg/mL, respectively) to digest the cell wall. Protoplasted cells were then subjected to an overnight enzymatic lysis in 1% Na-Lauroyl sarcosine buffer with Proteinase K at 5 mg/mL final concentration. The plugs were washed three times with TE buffer and digested overnight with beta-Agarase I (final concentration of 40 units/mL) to release the DNA into the solution. The gDNA extract was purified using the High salt-Phenol:Chloroform:IsoAmyl Alcohol protocol. Cell extracts were suspended in isoamyl alcohol in a high salt buffer, purified in subsequent steps with Phenol: Chloroform:Isoamyl Alcohol and Chloroform:Isoamyl Alcohol reagents, followed by precipitation of the gDNA with 100% ethanol. The resulting DNA pellet was washed twice with 70% Ethanol, dried and resuspended in TE buffer overnight at 4 °C. Purified gDNA was concentrated using AMPure PB beads. Qubit fluorometer and Agilent Bioanalyzer were used to determine DNA concentration and the insert size prior to sequencing. The protocol resulted in 8.3 μg of purified gDNA with an average size of 20–60 kb.

**Genome sequencing**. The *Scenedesmus* sp. NREL 46B-D3 genome was sequenced using PacBio technologies. Unamplified libraries were generated using PacBio standard template preparation protocol for creating >10 kb libraries (PacBio, Menlo Park, CA). The DNA was sheared using Covaris g-Tubes(TM) (Covaris Inc., Woburn, MA) to generate sheared fragments of >10 kb in length. The sheared DNA fragments were then prepared using Pacific Biosciences SMRTbell template preparation kit (Pacific Biosciences, Menlo Park, CA), where the fragments were treated with DNA damage repair, had their ends repaired so that they were blunt-ended, and 5' phosphorylated. Pacific Biosciences hairpin adapters (Pacific Biosciences, Menlo Park, CA) were then ligated to the fragments and exonuclease treated to remove failed ligation products and to create the SMRTbell templates for size-selection. The SMRTbell templates were then size-selected using the Sage Science BluePippin instrument (Sage Science, Beverly, MA) with a 6 kb lower cutoff resulting in an average library insert size of 15 kb. Pacific Biosciences sequencing primer (5′-AAAAAAAAAAAAAAAAAAAATATAACGGAGGAGGAGGA-3′; Pacific Biosciences, Menlo Park, CA) was then annealed to the SMRTbell templates and Version P6 sequencing polymerase was bound to them. The prepared SMRTbell template libraries were then sequenced on a Pacific Biosciences RSII sequencer (Pacific Biosciences, Menlo Park, CA) using Version C4 chemistry and 4-h sequencing movie run times.

**Genome assembly and annotation**. Filtered PacBio subread data was assembled together with Falcon v 0.7.3 to generate an initial assembly. Estimated fold coverage of the PacBio reads was 81.46 x. The mitochondrial and chloroplast assemblies were assembled with Celera and improved with finisherSC. The genome assembly was annotated using the JGI annotation pipeline.

During the annotation process, secondary scaffolds were detected by performing an all-against-all BLAT alignment[74]. If a pair of scaffolds were aligned with >50% coverage and >95% sequence identity, the longer scaffold was labeled as the primary scaffold and the shorter scaffold was labeled as the secondary scaffold. In all, 7,071 gene models on secondary scaffolds that fell within the alignment region between primary and secondary scaffolds were removed from the primary haplotype and excluded from the downstream analysis. In all, 1,606 of the 2,661 scaffolds are very similar to larger scaffolds and are predicted to constitute an alternate or secondary haplotype. Excluding the secondary scaffolds from the genome assembly size, the assembly size of the single haplotype is approximated to be 103.4 Mbp.

The assembly was masked for repeats using RepeatMasker (https://www.repeatmasker.org/), and the RepBase library[75], and the most frequent (>150) repeats recognized by RepeatScout[76]. Protein-coding gene models were predicted using the ab initio Fgenesh[77], homology-based Fgenesh+ and homology-based GeneWise[78] seeded by BLASTx alignments against the NCBI NR database, and transcriptome-based Fgenesh. Automated filtering selected the best model at each genomic locus based on homology and transcriptome support. Manual curation of transcription factor gene models was performed to obtain the final gene set, including gene models generated using Braker v1.9[79]. Predicted proteins were functionally annotated using SignalP[80] for signal sequences, TMHMM[81] for transmembrane domains, InterProScan[82], Priam[32] (http://priam.prabi.fr/, January 2018 release) for EC number assignment, and protein alignments to NCBI NR, SwissProt[83], KEGG[84], KOG[85] and TCDB for transporter classifications[30]. Hits from InterPro and SwissProt were used to map Gene Ontology (GO) terms[86]. Transcription factors were annotated using a curated set of Pfam domains and

using the PlantTFDB web server v 4.0[31] (http://planttfdb.cbi.pku.edu.cn). The completeness of the predicted protein-coding gene set was evaluated using BUSCO v 4.0.5 in protein mode on the Chlorophyta ortholog dataset (chlorophyta_odb10; 11-20-19)[87]. To assess the quality of the genome assembly, mapping of RNASeq reads to the genome assembly was performed using BLAT[74] with the thresholds of 95% nucleotide identity and 80% coverage over the read length. In addition to the nuclear genome, mitochondrial and chloroplast genomes were assembled separately. For the mitochondrial and chloroplast genomes, the RNA and protein-coding genes were annotated using the Chlorobox annotation web server GeSeq[88] (Supplementary Note 1, Supplementary Fig. 11). The current genome assembly and annotations are publicly available at the JGI Algal Genome Portal PhycoCosm[89]: https://phycocosm.jgi.doe.gov/Scesp_1 and were deposited to GenBank under accession JACERP000000000.

**Inference of species tree**. The phylogenetic tree was estimated from putative single-copy orthologs from Chlorophyta genomes identified using the software package OrthoFinder[90]. The genomes annotations were accessed from the JGI Algal Genome Portal PhycoCosm[89] (https://phycocosm.jgi.doe.gov): *Chlamydomonas reinhardtii* v5.6–Chlre5_6[91], *Chromochloris zofingiensis*–Chrzof1[92], *Coccomyxa* sp. C-169-Coc_C169_1[93], *Chlorella* sp. NC64A–ChlNC64A_1[94], *Dunaliella salina*–Dunsal1[95], *Gonium pectorale*-Gonpec1[96], *Monoraphidium neglectum*-Monneg1[97], *Raphidocelis subcapitata*-Rapsub1[98], *Scenedesmus obliquus* UTEX 393 (also, called *Tetradesmus obliquus* UTEX 393)-Sobl393_1[24], *Scenedesmus obliquus* UTEX 3031 (a subclone of DOE0152Z)-Sceobl1[28], and *Volvox carteri* v 2.1 – Volca2_1[99]. Using 606 single-copy orthologs, the trimmed alignments were concatenated into a single alignment, and the phylogenetic tree was inferred under maximum-likelihood (RAxML v 8.2.8) using automated bootstrapping, which converged after 100 bootstrap replicates (arguments: -f a -m PROTGAMMAWAGF)[100]. The tree was visualized using FigTree v 1.4.4 (http://tree.bio.ed.ac.uk/software/figtree/).

**Prediction of homologous gene families and expansion/contraction**. Homologous gene families were predicted using the OrthoFinder software package v 2.3.8[89]. The expansion and contraction of gene families were estimated using the Computational Analysis of gene Family Evolution (CAFE) software package v 4.2.1, which provides a statistical foundation for evolutionary inferences[101,102]. Gene families with >100 genes in a single genome were removed when estimating the birth-death parameter to ensure convergence. Expansions or contractions of gene families with a *p*-value ≤ 0.01 were considered significant, indicating that the observed gene family counts deviated from the expected model for gene gain and loss.

**RNA extraction and sequencing**. RNA was extracted, isolated, and sequenced from 36 samples. Extractions were performed using the Omega E.Z.N.A. Plant RNA Kit (Omega Bio-Tek, Norcross, GA) following manufacturer's protocols with the addition of three rounds of liquid-nitrogen freeze, thaw, and grinding for cell lysis[103]. The concentration of total RNA was obtained using the Qubit RNA BR Assay Kit (Thermo Fisher Scientific, Waltham, MA). The quality of the RNA was determined by the Agilent RNA 6000 Pico Kit (Agilent Technologies, Santa Clara, CA). Five micrograms of total RNA was depleted of ribosomal RNA using the Ribo-Zero H/M/R kit (Illumina Inc., San Diego, CA) and eluted in nuclease free water (Zymo Research, Irvine, CA). The concentration of the ribosomal depleted RNA was obtained using the Qubit RNA HS Assay (Thermo Fisher Scientific, Waltham, MA) and the quality was determined by the Agilent RNA 6000 Pico Kit (Agilent Technologies, Santa Clara, CA). The ribosomal RNA depleted samples were then processed using the NEBNext Poly(A) mRNA Magnetic Kit (New England Biolabs Inc., Ipswich, MA). The Poly(A) selected fraction was kept and the RNA concentration was obtained using the Qubit RNA HS Assay (Thermo Fisher Scientific, Waltham, MA) and the quality was determined by the Agilent RNA 6000 Pico Kit (Agilent Technologies, Santa Clara, CA). Illumina libraries were prepared using ScriptSeq v2 Library Preparation Kit (Illumina Inc., San Diego, CA). RNA was converted to cDNA and adapters and indexes were added onto the ends of the fragments to generate Illumina libraries. Illumina libraries were eluted in DNA Elution Buffer (Zymo Research, Irvine, CA), and the concentration was obtained using the Qubit double-stranded DNA (dsDNA) HS Assay (Thermo Fisher Scientific, Waltham, MA). The average size of the library was determined by the Agilent High Sensitivity DNA Kit (Agilent Technologies, Santa Clara, CA). An accurate library quantification was determined using the Library Quantification Kit –Illumina/Universal Kit (KAPA Biosystems Inc., Wilmington, MA). All libraries were multiplexed and sequenced on a NextSeq 500 to generate paired-end 151 bp reads using the NextSeq 500/550 High Output Kit v2.5 Kit (300 cycles) (Illumina, Cat. #20024908).

**Transcriptomic analysis**. Following Illumina sequencing, resulting demultiplexed forward and reverse fastq files were filtered and trimmed using the JGI QC pipeline. Using BBDuk, raw reads were evaluated for artifact sequence by kmer matching (kmer = 25), allowing one mismatch and detected artifact was trimmed from the 3' end of the reads. RNA spike-in reads, PhiX reads and reads containing

any Ns were removed. Quality trimming was performed using the phred trimming method set at Q6. Finally, following trimming, reads under the length threshold were removed (minimum length 25 bases or 1/3 of the original read length—whichever is longer). Filtered reads from each library were aligned to the reference genome using HISAT2 v 2.1.0. Strand-specific coverage bigWig files (fwd and rev) were generated using deepTools v 3.1, and featureCounts was used to generate the raw gene counts. DESeq2 (v 1.18.1)[104] was subsequently used to determine which genes were differentially expressed between pairs of treatment and control conditions at corresponding time points (cold vs. control and heat vs. control). A gene is considered differentially expressed between conditions when the adjusted *p*-value < 0.05 and |Log₂FC| > 1.

The gene co-expression network was calculated across expression profiles for the 6 time points across four different conditions (cold stress, heat stress, and two controls) using the R package WGCNA[34]. After filtering genes out due to low expression across > 95% of all conditions, 7,028 genes were used in correlation analysis. The correlations were scaled using soft power of 7, assuming a scale-free network. Hierarchical clustering was applied to identify co-expressed gene modules with a minimum cluster size of 30 genes. The network was visualized using Cytoscape. The scaled correlations between gene pairs above 0.1 are represented as edges in the network (Supplementary Data File 10). Functional enrichment for the gene clusters was performed using Gene Ontology assignments. The p-values for enrichment were calculated by the hypergeometric test and adjusted for multiple hypothesis testing using the Bonferroni correction.

**Metabolite extraction**. Prior to extraction, algae biomass (~20 mg) was first lyophilized dry (FreeZone 2.5 Plus, Labconco) then powderized by bead-beating with a 3.2 mm stainless steel bead in a bead-beater (Mini-Beadbeater-96, BioSpec Products) for 5 s (2x).

To extract lipids, a chloroform-based lipid extraction was performed using a modified Bligh-Dyer approach[105]. Here, 300 μL MeOH, 150 μL CH₃Cl and 120 μL water was added to the powderized pellet (final ratio of 2:1:0.8 MeOH:CH₃Cl:H₂O), vortexed and sonicated in a water bath for 10 min, then an additional 150 μL each of CH₃Cl and MeOH was added (final ratio of 1:1:0.9 MeOH:CH₃Cl:H₂O), which was followed by a brief vortex and 10 min incubation in a sonic water bath. Samples were centrifuged for 5 min at 5,000 rpm, and the bottom lipid-enriched chloroform phase was transferred to a new tube. An addition of 300 μL of chloroform was added, followed by another vortex, sonication, and centrifugation. The bottom chloroform phase was then combined with the previously collected extract. Lastly, chloroform extracts of lipid were dried in a SpeedVac (SPD111V, Thermo Scientific) and stored at −20 °C.

To extract polar metabolites, 450 μL methanol was added to powderized pellets, vortexed and sonicated in a water bath for 10 min, stored at −20 °C for 2 h, and finally vortexed and sonicated for 10 min. After centrifuging for 5 min at 5,000 rpm, 200 μL of supernatant was transferred to a separate tube. Extracts were then dried in a SpeedVac (SPD111V, Thermo Scientific) and stored at −80 °C.

**Liquid chromatography tandem mass spectrometry (LC-MS/MS)**. LC-MS/MS was performed on extracts of algae biomass using both normal and reverse phase chromatography for polar and lipid metabolites, respectively. Chromatography was performed using an Agilent 1290 LC stack, with MS and MS/MS data collected using a Q Exactive Orbitrap MS (Thermo Scientific, San Jose, CA).

Extracted lipids from 20 mg wet algae biomass were resuspended in 240 μL 3:3:4 IPA:ACN:MeOH containing a 4 μM internal standard mixture of deuterium-labeled lipids (Lyso PC (17:1)-#110905, PG-d5 (16:0/18:1)-#110899, DGTS-d9 (16:0/16:0)-#857463P, oleic acid (18:1-d9)-#861809O, PS (16:0-d31/18:1)-#110922, PA (16:0-d31/18:1)-#110920, PE (16:0-d31/18:1)-#110921, PC (16:0-d31/18:1)-#110918, DG-d5 (1,3-16:1)-#110579, TG-d5 (17:0/17:1/17:0)-#110544; Avanti Polar Lipids) and 1 μg/mL 2-Amino-3-bromo-5-methylbenzoic acid (Sigma, M9508). One-hundred twenty microliters of this was centrifuge-filtered through a 0.22 μm hydrophilic PVDF membrane (#UFC30GV00, Millipore) and transferred to a glass LC-MS vial. Full MS spectra were collected from *m/z* 80 to 1200 at 70,000 resolution in both positive and negative ion mode, with MS/MS fragmentation data acquired using stepped 10, 20, and 40 eV collision energies at 17,500 resolution. Chromatography was performed using a C18 column (Agilent ZORBAX Eclipse Plus C18, Rapid Resolution HD, 2.1 × 50 mm, 1.8 μm) at a flow rate of 0.4 mL/min with a 2 μL injection volume. To detect lipids, samples were run on the C18 column at 55 °C equilibrated with 100% buffer A (60:40 H₂O:ACN with 5 mM ammonium acetate and 0.1% formic acid) for 1 min, diluting buffer A down to 45% with buffer B (90:10 IPA:ACN w/ 5 mM ammonium acetate and 0.1% formic acid) over 2 min, down to 20% A over 8 min, then down to 0% A over 1.5 min, followed by isocratic elution in 100% buffer B for 1.5 min. Samples consisted of three biological replicates each and three extraction controls, with sample injection order randomized and an injection blank (2 μL of 3:3:4 IPA:ACN:MeOH) run between each sample.

Extracted polar metabolites from 25 to 105 mg wet algae biomass were resuspended in 100% methanol containing 10 μM internal standard (5–50 μM of 13C,15N Cell Free Amino Acid Mixture, #767964, Sigma), with the resuspension volume varied for each sample to normalize by biomass weight. One-hundred fifty microliters of this was centrifuge-filtered through a 0.22 μM hydrophilic PVDF

membrane (#UFC30GV00, Millipore) and transferred to a glass LC-MS vial. Full MS spectra were collected from $m/z$ 70 to 1,050 at 70,000 resolution in both positive and negative mode, with MS/MS fragmentation data acquired using stepped 10, 20, and 40 eV collision energies at 17,500 resolution. Chromatography was performed using a HILIC column (Agilent InfinityLab Poroshell 120 HILIC-Z, $2.1 \times 150$ mm, 2.7 μm, #673775-924) at a flow rate of 0.4 mL/min with a 2 μL injection volume. To detect metabolites, samples were run on the HILIC column at 40 °C equilibrated with 100% buffer B (95:5 ACN:H$_2$O with 5 mM ammonium acetate) for 1 min, diluting buffer B down to 89% with buffer A (100% H$_2$O with 5 mM ammonium acetate and 5 μM methylenediphosphonic acid) over 10 min, down to 70% B over 4.75 min, then down to 20% B over 0.5 min, followed by isocratic elution in 80% buffer A for 2.25 min. Samples consisted of three biological replicates each and three extraction controls, with sample injection order randomized and an injection blank (2 μL MeOH) run between each sample.

**Metabolomic identification**. Metabolites were identified based on exact mass and comparing retention time (RT) and MS/MS fragmentation spectra to that of standards run using the same chromatography and MS/MS method. LC-MS data was analyzed using custom Python code[106]. Each detected feature (unique m/z coupled with RT) was assigned a score (0 to 3) representing the level of confidence in the metabolite identification. Metabolites given a positive identification had detected m/z ≤ 5 ppm or 0.001 Da from theoretical as well as RT ≤ 0.5 min compared to a pure standard run using the same LC-MS method. A compound with the highest level of positive identification (score of 3) also had matching MS/MS fragmentation spectra compared to either an outside database (METLIN)[107] or internal database generated from standards run and collected on a Q Exactive Orbitrap MS. MS/MS mismatches to the standard invalidated an identification.

For a detected lipid compound, lipid class was determined based on characteristic fragment ions or neutral loss, and coupled with exact mass to determine specific lipid identity (number of carbons in fatty acid tails and degree of unsaturation). In POS mode, MGDG and DGDG lipids ionized as $[M+ NH4]+$ with a neutral loss of 179 and 341[108], respectively, and TGs as $[M+ NH4]+$ with fatty acid tails detected in the MS/MS spectra[109]. Lipid standards run separately (DGDG (18:3/18:3)-#840524 P, MGDG (18:3/16:3)-#840523 P; Avanti Polar Lipids, Inc.) and deuterated internal standards of DG and TG were used to verify fragmentation pattern and elution times of each lipid class. Owing to lipid degradation over total sample runtime, detected intensity for peak height was normalized to the deuterated DG internal standard for MGDG and DGDG lipids, and to the deuterated TG internal standard for TG lipids. Mass spectrometry results of identified metabolites are summarized in Supplementary Data File 13 and 14.

Untargeted analysis was performed using a Feature-Based Molecular Networking workflow[110] (version release 18) using MZmine2[111] and Global Natural Products Social Molecular Networking (GNPS)[33], a web-based mass spectrometry identification tool. An MZmine workflow was used to generate a list of features (mzrt values obtained from extracted ion chromatographs containing chromatographic peaks within a narrow m/z range) and filtered to remove isotopes, adducts and features without MS/MS. For each feature, the most intense fragmentation spectrum, was uploaded to the GNPS web tool, and a putative identification is made when a sample spectrum matches on deposited within the GNPS database. The molecular networking jobs can be publicly accessed at https://gnps.ucsd.edu/ProteoSAFe/status.jsp?task=e971a91e4ebd489f851a915c08d19ced for the positive ion mode dataset and https://gnps.ucsd.edu/ProteoSAFe/status.jsp?task=6b7d66e2b1cd487d9d8c5f345435b8f8 for the negative ion mode dataset.

**Statistics and reproducibility**. Compositional analysis used two biological replicates, as the range of variables tested and time points sampled were prioritized over additional replicates, reflecting previous methodologies. Samples used in transcriptomic and metabolomic analysis were collected every 4 h over 24 h for three biological replicates. For differential expression analysis, the DESeq2 (v 1.18.1)[104] software applied significance testing using the Wald test and Benjamini–Hochberg adjustment. For pathway enrichment, significance testing was performed using a one-sided hypergeometric test with the Bonferroni correction. For fold change in metabolomics measurements, significance testing was performed using a two-sided $t$-test.

**Reporting summary**. Further information on research design is available in the Nature Research Reporting Summary linked to this article.

## Data availability

The *Scenedesmus* sp. NREL 46B-D3 strain is available from the culture collection at National Renewable Energy Laboratory by contacting Michael Guarnieri. The current genome assembly and annotations are publicly available at the JGI Algal Genome Portal PhycoCosm[90]: https://phycocosm.jgi.doe.gov/Scesp_1. This Whole Genome Shotgun project has been deposited at DDBJ/ENA/GenBank under the accession JACERP000000000. The version described in this paper is version JACERP010000000. Genome sequencing data have been deposited to SRA under the accession PRJNA407785. Transcriptomics data have been deposited to SRA under the accession PRJNA672938.

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

## Acknowledgements

Special thanks to Juergen Polle for helpful discussions, Alan Kuo for assistance with genome annotation, Asaf Salamov for assistance with genome analysis and annotation, and Vasanth Signan for assistance with differential expression analysis. This research was supported by the Bioenergy Technology Office (BETO) within the Department of Energy's Office of Energy Efficiency and Renewable Energy (EERE) under Agreements NL0032355 and NL0032266. This work was also completed in part by the National Renewable Energy Laboratory, operated by Alliance for Sustainable Energy, LLC, for the U.S. Department of Energy (DOE) under Contract No. DE-AC36-08GO28308. The work conducted herein by the U.S. Department of Energy Joint Genome Institute, a DOE Office of Science User Facility, is supported by the Office of Science of the U.S. Department of Energy under Contract No. DE-AC02-05CH11231.

## Author contributions

S.S. and I.V.G. designed and supervised the study, secured funding, and edited the manuscript. C.D., D.D., and S.M. coordinated genome sequencing. K.La. assembled genome. K.Lo., A.K., D.T., B.B., T.N. produced metabolomics data. S.C. annotated genome, performed integrative multi-omics analysis, comparative genomics, wrote paper with contributions from T.A.S.B., L.R.D., M.T.G., S.S., I.V.G. L.R.D. downselected and characterized the strain under supervision and funding provided by M.T.G. Y.K. completed the genomic DNA extraction and contributed text for the manuscript. T.A.S.B. established and optimized photobioreactor cultivation, executed experiments and sample acquisition including RNA extraction and Q.C., performed RNA-seq analysis, and wrote sections of the manuscript.

## Competing interests

The authors declare no competing interests.
