## [Peer Review File · Communications Biology]

Reviewers' comments:

Reviewer #1 (Remarks to the Author):

In this manuscript, Calhoun et al. present a multi-omics approach integrating genome, transcriptome and metabolome data to characterize cold stress and heat stress in a novel halotolerant green algal strain, *Scenedesmus* sp. NREL 46B-D3. This algal strain is a potential candidate for biotechnological applications due to its fast growth rate, and capacity to accumulate secondary metabolites. The overall study involves an extensive amount of work, data generation, experimental design and the interpretation of the results appear reasonable (but see my comments below). The adopted methodology is largely industrial standard and is scientifically sound. The manuscript is well-written.

Main concerns:

1. Inconsistency of the number of replicates in the experimental design. While most of the key results are based on a 3 biological replicates (the general minimum requirement to gauge statistical significance), some are based on only 2 replicates (e.g. data presented in Fig. 1c, Fig 2; some data in Supp Figs 1 and 2). Much of the analysis based on 2 replicates did not attempt to assess statistical significance, and thus this does not necessarily indicate critical flaws in the design. While I do not see the need to repeat these experiments, the rationale behind such an experimental design should be at least justified or clarified in the text.

2. Genome assembly and completeness. The assembled genome (151.90 Mbp) is said to be diploid (page 6), with a "significant separation of alleles", but no detail is provided as to how this conclusion was reached. Was this done using GenomeScope2, Haplomerger and/or purge_haplotigs? How did you generate the haplotype? This should be clarified perhaps as a new subsection in Methods. Does the 151.90 Mbp represents the assembly size for a diploid genome, or the haplotype genome representation?

The use of BUSCO genes for assessing completeness of eukaryote genome assembly is common. Based on the text (page 6, top line), I suspect this was done using the transcriptome mode, but BUSCO analysis is not based on mapping of RNA-Seq reads, as the text suggests. Transcriptome mode is used to assess completeness of assembled transcripts (not RNA-Seq reads). If the assembled transcripts were used, would these be a reference transcriptome dataset (combining various sets generated from various conditions in this study)? If the purpose is to assess completeness of the genome assembly, an assessment using genome scaffolds (i.e. requiring TBLASTN on BUSCO) would be more appropriate; this may be the case based on the legend of Supp. Fig. 3. You may also use the protein mode to assess the completeness of the protein-coding genes predicted from the genome. However, no details of BUSCO analysis are provided in the Methods (in page 27). These details are necessary to interpret the results.

Minor comments (page numbers follow the submitted Word document):

1. Of the large number of figures, I find Figure 4 particularly confusing. The legend and the main text describes six sampling points within 24 hours (a sample every 4 hours, for transcriptome and metabolomes), while five data points (each at a different day, showing growth) are plotted in the graph for each of the four experiments. Some of the text in the legend is perhaps better presented in the Methods section, so the whole experimental set up can be further clarified.

2. For clarity, the different panels of each main figure should be cited in the text at appropriate locations. For instance, the three panels in Fig. 1 show very different information. The sentence in

page 4 citing "Figure 1" is misleading, as it suggests that Figure 1 shows the image/info about *Nannochloropsis salina* CCMP 1776, which is incorrect. Similarly, the citation of Figure 1 about the 9-day growth comparison in page 5 should really be Figure 1c; also, this info is presented after Figure 3. Consider presenting these graphs in Fig. 1c as Fig. 2d, or as a Supplementary Figure, since these graphs were only briefly mentioned in the text.

3. Description of differentially expressed genes should be clarified either in the Results or in the Methods. It should be clear the difference is relative to the control, and that a difference is considered significant at $p\text{-value} < XXX$. Given the rich information presented, e.g. in Figs 5 and 6, the asterisk on a heatmap, when not defined clearly, could be misinterpreted as difference between cold versus heat samples, or between "Heat 20h" versus "Heat 24h", instead of cold versus control, and heat versus control.

4. Page 3, 3rd line from bottom (and elsewhere e.g. pages 18, 22): "sp." should not be italicised.

5. Fig. 3 is perhaps better presented in multiple panels (a, b, c etc.). In Figure 3 and Supp. Fig. 4, bootstrap values should be shown on the maximum likelihood tree. Branch length unit should be described in the legend – I suspect the unit is in number of substitutions per site?

6. Page 6, "carteri,)" -> "carteri)"

7. Page 6 (bottom): 3,669 families are noted as single-copy gene families that are "unique to NREL 46B-D3". Does this mean that they are singletons (i.e. each of 3669 families has only 1 gene sequence from NREL 46B-D3)? Similarly, the 275 multi-copy gene families are unique to this strain; are these exclusively paralogs (i.e. each family has multiple genes only found in this strain)?

8. Page 7: "PFam" -> "Pfam"

9. Page 7, bottom line: "CIC-1" should be "ClC-1" (the second character is small letter L, not capital I; Cl for chloride)

10. Page 8: "Sphaeropleales" – I believe that the order name should not be italicised.

11. Page 10, bottom two paragraphs: the presentation of results in the text (aa, other aa, polyamines, and CCM) does not follow the order of the heatmaps presented in Fig. 6a. Also, Fig. 6b is never cited.

12. Page 11, "In contrast, the heat stress samples 2,168 20% of total genes)." – this sentence is incomplete.

13. Fig. 1a is cited as the "pigmentation observed in the micrograph" in page 11. This detail is missing in the figure legend. Perhaps annotate the figure (e.g. using arrows) indicating the pigments and mention this in the legend?

14. Page 12: "module is colored tan in Figure 7C" – this is very difficult to figure out, as the colour "tan" is not easy to discern. Similarly, in the bottom line, "module is colored blue in Figure 7C" – there are at least three shades of blue in the figure across the three networks, which one is this statement referring to? Would it be possible to provide a legend specific these described modules and annotate in the figure?

15. Page 14 (bottom): The gene names presented in the text are not directly relatable to the information presented in Supp Fig. 9a.

16. Page 20: Targeted modulation "of" or "in"?
17. Page 23: in the last sentence of Conclusions, "NREL" is missing in the strain name. Bottom line (and top page 24), "bottom panel of Figure 1" -> "Fig. 1c"? Also, figures (e.g. "Figure 2") would not utilize salinity.
18. Page 24: "optima" (plural) or "optimum" (singular)?
19. Page 25: does the culturing condition follows earlier studies or based on some sort of optimisation? I find the 18h:6h light:dark cycle and the precise temperature of 28.2 degree Celsius interesting, and I wonder if the authors could provide a brief justification or clarification of this approach, or cite the relevant paper(s). For instance, the heat setting (33.2 C) is exactly 5 degrees above the control of 28.2, but the cold setting (13 C) is 15.2 C lower than the control, which is not directly relatable to what was described at the bottom of page 24.
20. Page 25: final concentration of some reagents/chemicals is not provided in the DNA extraction protocol, e.g. concentration of Proteinase K, beta-agarase I etc.
21. Top of page 26: I think the term "QC'd" should be avoided in the text. The fragment sizes of gDNA should be described in (kilo)bases, e.g. "60kb", nor "60k".
22. Genome sequencing: what is the selected size-range for the DNA fragments? 10-20Kb?
23. Page 28: the subheading "Tree building" may be better represented as "Inference of species tree"
24. Page 28 (in "Prediction of orthologous gene families ..."): the "orthogroups" in the output of OrthoFinder do not necessarily represent orthologous gene families (i.e. families of genes that arisen via speciation event). Unless these are single-copy gene/protein families (i.e. the 606 families you used to infer the species tree), all other so-called "orthogroups" would contain paralogs (such as the families with >100 genes from a single genome described here). For clarity, the better and more-inclusive term is "homologous" gene families.
25. Pages 29-30: on which Illumina platform was the RNA-Seq data generated? HiSeq 4000 or NovaSeq 6000? The platform used would affect the tools used for analysis, e.g. NovaSeq 6000 data tend to have polyG sequences.
26. Fig. 6, Supp. Fig. 7 – number of replicates is unclear. In Fig. 6b, it is unclear that the bidirectional horizontal arrow (top left) indicates "no change". I first misinterpreted it as a reversible chemical reaction involving L-Glutamate (and that the other compound is missing from the figure). It may be clearer if the figure is annotated with "no change" in this instance.
27. Page 33, top line: I believe you meant 0.22 micrometre (unit with small letter m), not micromolar.

Reviewer #2 (Remarks to the Author):

The manuscript under review gives novel and interesting results. However, the whole study is not very forthright and readers become in several points confused. From a technical point of view, it is a well conducted study (many state-of-the-art analyses). However, there are several points on the

presentation and on the experimental design that need more explanation and could be improved. The part of the strain selection is not well presented and needs more clarification on how the parameters were varied (salinity, N/P ration etc.). For the sake of the clarity of the manuscript this part could be either moved to the supplementary section or completely removed since it does not add any significant value on the main body of the study.

It is not clear how the nutrient deprivation is related to the core experiments of the study (to follow up omics changes in temperature stressed conditions). Please revise the introduction and in general the overall manuscript in order to give and discuss directly the main scientific questions of the study. The only real simulation of Arizona conditions was the light intensity. The question is: does the temperature of a culture drops at the rate you did applied (more or less instantly from 28 to 13 oC; Is this drop observed in real cultures?). So, one of the main problems of the manuscript is that it needs more clarity on the main scientific hypotheses set. The experimental design needs more analysis and justification regarding the selection of the parameters (temperature levels, temperature drop rate, time of cultivation etc.). So, it is not very clear what is the main scientific hypothesis (besides to follow the omics changes in temperature stressed cultures). Is it to follow the instant drop of temperature from 28 to 13 (or the increase to 32) and why, or to follow the changes during acclimation to lower or higher temperatures? In case of acclimation, I doubt that 24 hours were enough for acclimation.

Authors need to define the term "stress" used and subsequently the term "acclimation".

Also, I found that the term "post cell division" is used in a peculiar way.

Reviewer #3 (Remarks to the Author):

This manuscript employed multi-omic approaches to characterize temperature stress in a novel halotolerant microalga *Scenedesmus* sp. NREL 46B-D3. Under its optimal growth temperature at 25oC, NREL 46B-D3 has the highest biomass accumulation capacity post cell division reported to date for a halotolerant strain. The authors monitored the growth of NREL 46B-D3 under varying temperatures, salinities, and nitrogen:phosphorus (N:P) ratios to further characterize optimal and limiting growth conditions. They analyzed the genome sequence of NREL 46B-D3 and performed metabolomic and transcriptomic analysis in NREL 46B-D3 cultures grown under control condition (28.2°C), or cold stress (13°C), or high temperature stress (33.2°C) for 24h. Their results showed that genes involved in lipid production, ion channels and antiporters are expanded and expressed in NREL 46B-D3. Additionally, they showed that temperature stress shifts fatty acid metabolism and increases amino acids synthesis in NREL 46B-D3. Using co-expression analysis, they also identified several transcription factors that may control many fatty acid biosynthesis genes.

The research presented in this paper will have significant impact on algal biotechnology and algal biofuel production, considering the high biomass production in NREL 46B-D3. The multi-omic approaches they employed can provide useful information not only for the characterization of NREL 46B-D3 but also for other algal related research. Their thorough analysis of metabolomics, especially lipids, was quite impressive and presented very well using heatmaps.

I have some suggestions/comments about this paper.

Main comments:

1. For temperature stress experiment, why did the authors use these temperatures: control (28.2°C), cold (13 °C), or heat (33.2 °C)? The cold stress seems much stronger than the heat stress. Why was 28.2oC used as control? The optimal temp for NREL 46B-D3 is 25oC. What light intensity was used in the temperature stress experiment?
2. The paper starts with NREL 46B-D3 has higher biomass, and then talks about the temperature

stress. The link between the higher biomass and the temperature stress is not very clear, since the high biomass was produced under optimal temperature at 25°C.

3. The author characterized the optimal and limiting growth conditions for NREL 46B-D3, e.g. salinity, N:P ratio. Light is also an important factor for algal cultivation but has never been mentioned in the paper.

4. The result that NREL 46B-D3 produces more biomass than other strains is very interesting. But the reason for that is not clear. Does NREL 46B-D3 grow faster and/or have higher photosynthetic rates (e.g. O₂ evolution rate) or higher tolerance to stresses than other strains?

Minor comments:

1. Fig 1a, are the NREL 46B-D3 cells always in the clumping status?

2. Does Fig 1b have error bars? Why not including UTEX 393 in Fig 1b?

3. Fig 1C, the data is presented as g/L, what was the cell density difference between these two strains? Did NREL 46B-D3 grow faster than UTEX 393 or does NREL 46B-D3 have more biomass per cell? Do you have OD750 curves for NREL 46B-D3 and UTEX 393 for comparison?

4. Fig.4, why was there a drop of OD750 at 2day control cold?

5. Fig 7C is confusing. It is difficult to understand the first group and 2nd group mentioned in the paper.

6. Supplementary Fig. 6, it would be more elegant if putting the log₂FC color bar by the side of the heatmap, but not on the top.

7. "Thirty-five up-regulated and 46 down-regulated TFs were differentially expressed exclusively in cold stress samples." Is this for Fig 7b? If so, the numbers don't match the figure.

8. "Changes in the expression of genes involved in fatty acid synthesis were only observed in the cold stress but not the heat stress, suggesting that fatty acid synthesis is only modulated in *Scenedesmus* sp. NREL 46B-D3 in response to cold." I'm not sure if the statement is correct. It could also be because the heat stress was not strong enough to see the effects on the expression of genes involved in fatty acid synthesis.

9. The authors mentioned "General amino acid accumulation is a mechanism to relieve oxidative stress". Maybe have more discussion about this. What is the source of oxidative stress? What's the function of amino acids in tolerance of oxidative stress?

10. The strain selection condition is "temperature cycles from 21 to 32 °C, while lighting cycles from 0 to 965 μmol m⁻² s⁻¹." Maybe add some details of these cycles, how long for 21 or 32°C? Did the authors use direct temperature change or gradual temperature change? The omics data under 33.2°C and 13°C may not explain the increased biomass accumulation in this strain.

We thank you and the reviewers for reading our manuscript and providing insightful comments. We significantly modified the text of manuscript and the figures based on the reviewer suggestions. Below please find our responses to each of the comments, and we also highlighted the modifications in the text of the revised manuscript.

Reviewers' comments:

Reviewer #1 (Remarks to the Author):

In this manuscript, Calhoun et al. present a multi-omics approach integrating genome, transcriptome and metabolome data to characterize cold stress and heat stress in a novel halotolerant green algal strain, *Scenedesmus* sp. NREL 46B-D3. This algal strain is a potential candidate for biotechnological applications due to its fast growth rate, and capacity to accumulate secondary metabolites. The overall study involves an extensive amount of work, data generation, experimental design and the interpretation of the results appear reasonable (but see my comments below). The adopted methodology is largely industrial standard and is scientifically sound. The manuscript is well-written.

Main concerns:

1. Inconsistency of the number of replicates in the experimental design. While most of the key results are based on a 3 biological replicates (the general minimum requirement to gauge statistical significance), some are based on only 2 replicates (e.g. data presented in Fig. 1c, Fig 2; some data in Supp Figs 1 and 2). Much of the analysis based on 2 replicates did not attempt to assess statistical significance, and thus this does not necessarily indicate critical flaws in the design. While I do not see the need to repeat these experiments, the rationale behind such an experimental design should be at least justified or clarified in the text.

We chose to prioritize sampling of additional time points and variables, instead of additional replication, as we were limited by the number of reactor vessels in our custom built photobioreactor. Prior publications by our team and others has indicated statistically meaningful data can be obtained from 2 biological replicates (e.g. Dahlin et al., *Communications Biology*, 2019, <https://doi.org/10.1038/s42003-019-0620-2> and Ajjawi et al., *Nature Biotechnology*, 2017, <https://doi.org/10.1038/nbt.3865>). Manuscript text on page 26 was updated to more accurately reflect replication, and direct the reader to our prior reports as follows:

“Data relating to Figure 1c, Figure 2, Supplemental Figure 1 b,c, and Supplemental Figure 2 utilized two biological replicates, as the range of variables tested and time points sampled were prioritized over additional replicates, reflecting previous methodologies^{3,71}.”

2. Genome assembly and completeness. The assembled genome (151.90 Mbp) is said to be diploid (page 6), with a “significant separation of alleles”, but no detail is provided as to how this conclusion was reached. Was this done using GenomeScope2, Haplomerger and/or purge_haplotigs? How did you generate the haplotype? This should be clarified perhaps as a

new subsection in Methods. Does the 151.90 Mbp represents the assembly size for a diploid genome, or the haplotype genome representation?

The 151.9 Mbp size is the diploid genome assembly size, and the estimated haploid genome size is 103.4 Mbp. We estimated the haplotype by detecting scaffold pairs that show high similarity. An all-by-all BLAT alignment is performed using all scaffolds. For a pair of scaffolds that are aligned with >50% coverage and >95% sequence identity, the longer scaffold is considered the primary scaffold and the shorter scaffold is considered the secondary scaffold. Gene models that fall within the alignment region between primary and secondary scaffolds were removed from the primary haplotype and excluded from the downstream analysis. The procedure for determining the haplotype is now described in the Methods section under “Genome assembly and annotation” as a separate paragraph on page 29 and shown below:

“During the annotation process, secondary scaffolds were detected by performing an all-against-all BLAT alignment⁷³. If a pair of scaffolds were aligned with >50% coverage and >95% sequence identity, the longer scaffold was labeled as the primary scaffold and the shorter scaffold was labeled as the secondary scaffold. 7,071 gene models on secondary scaffolds that fell within the alignment region between primary and secondary scaffolds were removed from the primary haplotype and excluded from the downstream analysis. 1,606 of the 2,661 scaffolds are very similar to larger scaffolds and are predicted to constitute an alternate or secondary haplotype. Excluding the secondary scaffolds from the genome assembly size, the estimated genome size of the single haplotype is 103.4 Mbp.”

The use of BUSCO genes for assessing completeness of eukaryote genome assembly is common. Based on the text (page 6, top line), I suspect this was done using the transcriptome mode, but BUSCO analysis is not based on mapping of RNA-Seq reads, as the text suggests. Transcriptome mode is used to assess completeness of assembled transcripts (not RNA-Seq reads). If the assembled transcripts were used, would these be a reference transcriptome dataset (combining various sets generated from various conditions in this study)? If the purpose is to assess completeness of the genome assembly, an assessment using genome scaffolds (i.e. requiring TBLASTN on BUSCO) would be more appropriate; this may be the case based on the legend of Supp. Fig. 3. You may also use the protein mode to assess the completeness of the protein-coding genes predicted from the genome. However, no details of BUSCO analysis are provided in the Methods (in page 27). These details are necessary to interpret the results.

BUSCO was run in protein mode to assess the completeness of the genome annotation, not based on the mapping of RNASeq reads. We revised the text in the Results to clarify the BUSCO analysis (page 7):

“The completeness of the genome annotation was estimated to be 93.1% complete based on Chlorophyta BUSCO families (chlorophyta_obd10; 11-20-19)^{26,27}”

In addition, we revised the sentence on the BUSCO analysis in the Methods as follows:

“Completeness of the genome annotation was evaluated using BUSCO v 4.0.5 in protein mode on the Chlorophyta ortholog dataset (chlorophyta_obd10; 11-20-19)”.

RNA mapping to the genome was performed separately as an assessment of genome assembly quality, and a sentence describing the RNA mapping was added to the Methods (page 30):

“To assess the quality of the genome assembly, mapping of RNASeq reads to the genome assembly was performed using BLAT⁷³ with the thresholds of 95% nucleotide identity and 80% coverage over the read length.”

Minor comments (page numbers follow the submitted Word document):

1. Of the large number of figures, I find Figure 4 particularly confusing. The legend and the main text describes six sampling points within 24 hours (a sample every 4 hours, for transcriptome and metabolomes), while five data points (each at a different day, showing growth) are plotted in the graph for each of the four experiments. Some of the text in the legend is perhaps better presented in the Methods section, so the whole experimental set up can be further clarified.

We made changes to Figure 4 to more clearly show the sampling points over 24-hours within the context of growth over 5 days. The text from the legend that repeated experimental set-up details already presented in the methods was removed.

2. For clarity, the different panels of each main figure should be cited in the text at appropriate locations. For instance, the three panels in Fig. 1 show very different information. The sentence in page 4 citing “Figure 1” is misleading, as it suggests that Figure 1 shows the image/info about *Nannochloropsis salina* CCMP 1776, which is incorrect. Similarly, the citation of Figure 1 about the 9-day growth comparison in page 5 should really be Figure 1c; also, this info is presented after Figure 3. Consider presenting these graphs in Fig. 1c as Fig. 2d, or as a Supplementary Figure, since these graphs were only briefly mentioned in the text.

We have altered the in-text Figure citations to reference the correct panel for each figure.

3. Description of differentially expressed genes should be clarified either in the Results or in the Methods. It should be clear the difference is relative to the control, and that a difference is considered significant at $p\text{-value} < XXX$. Given the rich information presented, e.g. in Figs 5 and 6, the asterisk on a heatmap, when not defined clearly, could be misinterpreted as difference between cold versus heat samples, or between “Heat 20h” versus “Heat 24h”, instead of cold versus control, and heat versus control.

We agree that this is an important point for interpreting the differential expression analysis. We revised the text in the Results under the first paragraph of Transcriptomic Response (page 12) and the Methods under Transcriptomic analysis (page 33) to emphasize this point:

“DESeq2 (v 1.18.1) was subsequently used to determine which genes were differentially expressed between pairs of treatment and control conditions at corresponding time points (cold

vs. control and heat vs. control). A gene is considered differentially expressed between conditions when the adjusted p-value < 0.05 and $|\text{Log}_2\text{FC}| > 1$.”

4. Page 3, 3rd line from bottom (and elsewhere e.g. pages 18, 22): “sp.” should not be in italicised.

We fixed the formatting where there was incorrect italicization in the text and in Supplementary Figure 3.

5. Fig. 3 is perhaps better presented in multiple panels (a, b, c etc.). In Figure 3 and Supp. Fig. 4, bootstrap values should be shown on the maximum likelihood tree. Branch length unit should be described in the legend – I suspect the unit is in number of substitutions per site?

We modified Figure 3 based on the reviewer's suggestions with letters indicating separate panels. For the maximum likelihood tree shown in Figure 3, bootstrap values were 100% for all branches, which is now mentioned in the figure legend. We added bootstrap values to branches on the tree in Supplementary Figure 4. Also, we added a note about the scale bar in both of the figure legends for Figure 3 and Supplementary Figure 4.

6. Page 6, “carteri,)” -> “carteri)”

We removed the extra comma.

7. Page 6 (bottom): 3,669 families are noted as single-copy gene families that are “unique to NREL 46B-D3”. Does this mean that they are singletons (i.e. each of 3669 families has only 1 gene sequence from NREL 46B-D3)? Similarly, the 275 multi-copy gene families are unique to this strain; are these exclusively paralogs (i.e. each family has multiple genes only found in this strain)?

The reviewer is correct that the unique (or orphan) gene families in the NREL 46-D3 genome include 3,669 singleton genes and 275 gene families made up of paralogs (936 individual genes).

8. Page 7: “PFam” -> “Pfam”

We edited all instances of “PFam” present in the text.

9. Page 7, bottom line: “CIC-1” should be “ClC-1” (the second character is small letter L, not capital I; Cl for chloride)

We fixed the typo for the Chloride ion channel.

10. Page 8: “Sphaeropleales” – I believe that the order name should not be italicised.

We agree with the reviewer and have fixed the formatting of the order name.

11. Page 10, bottom two paragraphs: the presentation of results in the text (aa, other aa, polyamines, and CCM) does not follow the order of the heatmaps presented in Fig. 6a. Also, Fig. 6b is never cited.

The text on page 11 describing the results of the metabolomics data and Figure 6A were revised such that the order of the results is consistent with the heatmaps in the figure. In addition, in-text references to Figure 6b were added on pages 11 and 18.

12. Page 11, “In contrast, the heat stress samples 2,168 20% of total genes).” – this sentence is incomplete.

We edited this sentence in the text on page 12 so that it reads as follows:

“By contrast, in the heat stress samples, 2,168 genes increased in expression (about 12% of total genes) and 3,460 genes decreased in expression (about 20% of total genes).”

13. Fig. 1a is cited as the “pigmentation observed in the micrograph” in page 11. This detail is missing in the figure legend. Perhaps annotate the figure (e.g. using arrows) indicating the pigments and mention this in the legend?

We have updated Figure 1 to include an arrow highlighting this pigmentation and have noted this in the figure legend.

14. Page 12: “module is colored tan in Figure 7C” – this is very difficult to figure out, as the colour “tan” is not easy to discern. Similarly, in the bottom line, “module is colored blue in Figure 7C” – there are at least three shades of blue in the figure across the three networks, which one is this statement referring to? Would it be possible to provide a legend specific these described modules and annotate in the figure?

We modified Figure 7c such that the two co-expression modules are clearly labeled and annotated in the network.

15. Page 14 (bottom): The gene names presented in the text are not directly relatable to the information presented in Supp Fig. 9a.

The text describing fatty acid metabolism genes in the Results section under “Transcriptomic response” and “Changes in expression of genes involved in fatty acid metabolism” (pages 13-17) has been revised to include the abbreviated genes names consistent with the labels in Supplementary Figure 9a.

16. Page 20: Targeted modulation “of” or “in”?

We edited the sentence, which was missing the word “of”.

17. Page 23: in the last sentence of Conclusions, “NREL” is missing in the strain name. Bottom line (and top page 24), “bottom panel of Figure 1” -> “Fig. 1c”? Also, figures (e.g. “Figure 2”) would not utilize salinity.

We have added the missing “NREL” to the strain name in the conclusions. We have also changed the “bottom panel of Figure 1” language to simply state Figure 1c. We have updated the manuscript text to indicate the salinity used in experiments relating to specific figures (page 26).

18. Page 24: “optima” (plural) or “optimum” (singular)?

We have corrected this typo to optimum.

19. Page 25: does the culturing condition follows earlier studies or based on some sort of optimisation? I find the 18h:6h light:dark cycle and the precise temperature of 28.2 degree Celsius interesting, and I wonder if the authors could provide a brief justification or clarification of this approach, or cite the relevant paper(s). For instance, the heat setting (33.2 C) is exactly 5 degrees above the control of 28.2, but the cold setting (13 C) is 15.2 C lower than the control, which is not directly relatable to what was described at the bottom of page 24.

We appreciate this comment and agree further clarification is warranted. The light:dark of 18:6 was selected to accommodate a sampling time point from the light period that immediately followed the dark cycle.

Temperatures were chosen to simulate the observed summer temperatures (May, June, and July, 2014) outdoor conditions of 1000 L algal ponds at the Arizona Center for Algae Technology and Innovation testbed site located in Mesa, Arizona. During the strain screening process, NREL 46B-D3 was up-selected for heat tolerance. We used the average daytime temperature, 28.2°C, as the control temperature in order to mimic actual observed temperatures from the 1000 L pond. Our selected control temperature was within range of the determined temperature optima (Supplementary Figure 1). Subsequently, the heat stress temperature, 33.2 °C, was the observed high, and the cold stress temperature, 13 °C, was the observed low from the same time period.

The text of the manuscript was modified accordingly to clarify (pages 26-27):

“In order to evaluate the effect of suboptimal outdoor temperature, *Scenedesmus* sp. NREL 46B-D3 was grown in summer temperatures (May, June, and July, 2014) that were observed during outdoor cultivation in Mesa, Arizona at the Arizona Center for Algae Technology and Innovation testbed site. The average afternoon temperature, 28.2 °C, was selected as the control temperature and within the determined optimal temperature range (Supplementary Figure 1). Subsequently, the heat stress, 33.2 °C, was the observed high temperature while the cold stress, 13 °C, was the observed low from the same time period.”

20. Page 25: final concentration of some reagents/chemicals is not provided in the DNA extraction protocol, e.g. concentration of Proteinase K, beta-agarase I etc.

Additional details for the extraction methodology were added to the text on pages 27-28.

21. Top of page 26: I think the term “QC’d” should be avoided in the text. The fragment sizes of gDNA should be described in (kilo)bases, e.g. “60kb”, nor “60k”.

The fragment size was specified, and text was revised to describe the quality control procedure on page 28.

22. Genome sequencing: what is the selected size-range for the DNA fragments? 10-20Kb?

The range of DNA fragments was 6-48kb with the average library insert size of 15kb.

23. Page 28: the subheading “Tree building” may be better represented as “Inference of species tree”

The subheading for “Tree building” was changed to “Inference of species tree” as suggested.

24. Page 28 (in “Prediction of orthologous gene families ...”): the “orthogroups” in the output of OrthoFinder do not necessarily represent orthologous gene families (i.e. families of genes that arisen via speciation event). Unless these are single-copy gene/protein families (i.e. the 606 families you used to infer the species tree), all other so-called “orthogroups” would contain paralogs (such as the families with >100 genes from a single genome described here). For clarity, the better and more-inclusive term is “homologous” gene families.

We agree with the reviewer that homologous is the more appropriate term to use for describing the set of genes identified through the Orthofinder analysis and have edited the text when this term is used.

25. Pages 29-30: on which Illumina platform was the RNA-Seq data generated? HiSeq 4000 or NovaSeq 6000? The platform used would affect the tools used for analysis, e.g. NovaSeq 6000 data tend to have polyG sequences.

Additional details on the sequencing platform were added to the text (page 33):

“All libraries were multiplexed and sequenced on a NextSeq 500 to generate paired-end 151 bp reads using the NextSeq 500/550 High Output Kit v2.5 Kit (300 cycles) (Illumina, Cat. #20024908).”

26. Fig. 6, Supp. Fig. 7 – number of replicates is unclear. In Fig. 6b, it is unclear that the bidirectional horizontal arrow (top left) indicates “no change”. I first misinterpreted it as a

reversible chemical reaction involving L-Glutamate (and that the other compound is missing from the figure). It may be clearer if the figure is annotated with “no change” in this instance.

We have added a note about the number of replicates in the figure captions for Figure 6 and Supplementary Figure 7. We also modified the arrow label for L-Glutamate in Figure 6b to minimize possible confusion.

27. Page 33, top line: I believe you meant 0.22 micrometre (unit with small letter m), not micromolar.

The typo was changed to micrometre in the text.

Reviewer #2 (Remarks to the Author):

The manuscript under review gives novel and interesting results. However, the whole study is not very forthright and readers become in several points confused. From a technical point of view, it is a well conducted study (many state-of-the-art analyses). However, there are several points on the presentation and on the experimental design that need more explanation and could be improved.

The part of the strain selection is not well presented and needs more clarification on how the parameters were varied (salinity, N/P ration etc.). For the sake of the clarity of the manuscript this part could be either moved to the supplementary section or completely removed since it does not add any significant value on the main body of the study.

To address the concern for more clarification on strain selection, we have added the following text to summarize details related to screening in the Results section (page 5):

“Briefly, 100 mL cultures were screened under the following conditions: 21-32°C temperature cycling, 35 g/L salinity, 0 to 965 $\mu\text{mol m}^{-2} \text{s}^{-1}$ light cycling, and constant 2% CO_2 sparging.”

Strain selection was performed as reported previously (Dahlin et al., *Communications Biology*, 2019, <https://doi.org/10.1038/s42003-019-0620-2>, and Dahlin et al., *Front. Plant. Sci.*, 2018, <https://doi.org/10.3389/fpls.2018.01513>). Furthermore, details are available in the Methods for how parameters were varied on pages 25 and 26.

It is not clear how the nutrient deprivation is related to the core experiments of the study (to follow up omics changes in temperature stressed conditions). Please revise the introduction and in general the overall manuscript in order to give and discuss directly the main scientific questions of the study.

The introduction was revised significantly to emphasize that the primary motivation for the study is the characterization of the systems level response to temperature stress in this high productivity strain (pages 3-5). For example, we added the following sentence to state the main objective of this study (page 4):

“Thus, the primary goals of this study were to first identify and characterize a high-productivity strain of interest, then subject it to a battery of multi-omics analyses to baseline its metabolic response to known environmental pressures to identify genetic targets and/or biochemical pathways that when modified, may enhance the stability and productivity of the algae in suboptimal cultivation conditions.”

Also, we revised the following sentence to clarify the motivation for this study (page 5):

“The resultant data lays the foundation for subsequent genetic manipulation, and strain enhancement¹⁷⁻²⁰ to improve robust outdoor cultivation at sub-optimal environmental temperatures.”

To focus the introduction on these goals, we removed the description of previous nutrient deprivation studies in the text. Instead, the revised introduction focuses on temperature as a primary environmental driver of algal productivity and the main interest of this study.

The only real simulation of Arizona conditions was the light intensity. The question is: does the temperature of a culture drop at the rate you did applied (more or less instantly from 28 to 13 oC; Is this drop observed in real cultures?). So, one of the main problems of the manuscript is that it needs more clarity on the main scientific hypotheses set. The experimental design needs more analysis and justification regarding the selection of the parameters (temperature levels, temperature drop rate, time of cultivation etc.). So, it is not very clear what is the main scientific hypothesis (besides to follow the omics changes in temperature stressed cultures). Is it to follow the instant drop of temperature from 28 to 13 (or the increase to 32) and why, or to follow the changes during acclimation to lower or higher temperatures? In case of acclimation, I doubt that 24 hours were enough for acclimation.

The text was modified to clarify the rationale for the selected temperature parameters (please see our response to comment #19 from the first reviewer). Heat and cold samples were cultivated at the control temperature before being moved into their respective growth chambers. While we did not measure the rate at which the temperature changed, we can roughly estimate that cultures reached their new temperature within the first few hours and well before sampling began reflecting the on average timing to achieve respective high or low temperature observed in the 1000 L ponds. This timing was selected in order to capture the sustained stress response rather than the initial shock.

We agree with the reviewer's comment regarding “acclimation” and address it below.

Authors need to define the term “stress” used and subsequently the term “acclimation”.

Referencing Fogg in “Algal Adaptation to Environmental Stresses” (2011, https://doi.org/10.1007/978-3-642-59491-5_1), we use “stress” to describe the initial distortion of

metabolic network processes caused by sudden temperature change and define it as such in our revisions (page 4), shown below.

“One of the primary environmental drivers of algal productivity is temperature. During mass cultivation in open ponds, fluctuations in temperature can stress or temporarily distort metabolic network processes affecting growth, overall productivity, and biomass composition.”

Here, we are using the term “acclimation” to refer to the first 24-hours of exposure to either the heat or cold temperature treatment. Upon a further consideration and review of the literature, we realize that the use of “acclimation” is not suitable in this context. Instead, this 24-hour window is now referred to as an “adjustment” period and reflected in our revisions.

Also, I found that the term “post cell division” is used in a peculiar way.

Based on the reviewer’s suggestion, we have updated the text to reflect our intent of separating the processes of cell division and biomass accumulation on page 4:

“Following cessation of cell division (stationary phase), this strain shows remarkable photosynthetic activity, concurrently accumulating new biomass almost exclusively as lipids and carbohydrates.”

Reviewer #3 (Remarks to the Author):

This manuscript employed multi-omic approaches to characterize temperature stress in a novel halotolerant microalga *Scenedesmus* sp. NREL 46B-D3. Under its optimal growth temperature at 25°C, NREL 46B-D3 has the highest biomass accumulation capacity post cell division reported to date for a halotolerant strain. The authors monitored the growth of NREL 46B-D3 under varying temperatures, salinities, and nitrogen:phosphorus (N:P) ratios to further characterize optimal and limiting growth conditions. They analyzed the genome sequence of NREL 46B-D3 and performed metabolomic and transcriptomic analysis in NREL 46B-D3 cultures grown under control condition (28.2°C), or cold stress (13°C), or high temperature stress (33.2°C) for 24h. Their results showed that genes involved in lipid production, ion channels and antiporters are expanded and expressed in NREL 46B-D3. Additionally, they showed that temperature stress shifts fatty acid metabolism and increases amino acids synthesis in NREL 46B-D3. Using co-expression analysis, they also identified several transcription factors that may control many fatty acid biosynthesis genes.

The research presented in this paper will have significant impact on algal biotechnology and algal biofuel production, considering the high biomass production in NREL 46B-D3. The multi-omic approaches they employed can provide useful information not only for the characterization of NREL 46B-D3 but also for other algal related research. Their thorough analysis of metabolomics, especially lipids, was quite impressive and presented very well using heatmaps.

I have some suggestions/comments about this paper.

Main comments:

1. For temperature stress experiment, why did the authors use these temperatures: control (28.2°C), cold (13 °C), or heat (33.2 °C)? The cold stress seems much stronger than the heat stress. Why was 28.2°C used as control? The optimal temp for NREL 46B-D3 is 25°C. What light intensity was used in the temperature stress experiment?

The text was modified to clarify the rationale for the selected temperature parameters (please see our response to comment #19 from the first reviewer). The cultures were grown under a light intensity of 300 $\mu\text{mol}/\text{m}^2/\text{s}$ to photosaturate the cultures. The light intensity value is now reported on page 26, first paragraph of "Temperature Challenges Studies" section of the Methods.

2. The paper starts with NREL 46B-D3 has higher biomass, and then talks about the temperature stress. The link between the higher biomass and the temperature stress is not very clear, since the high biomass was produced under optimal temperature at 25°C.

The high biomass production under its optimal temperature was the rationale for selecting the NREL 46B-D3 for characterization as a potential biofuel feedstock. Although the temperature stress experiments may not reveal why there is increased biomass accumulation, understanding the impact of temperature stress on productivity and the overall metabolic response in the NREL 46B-D3 is highly informative for strain improvement. As mentioned in response to the previous reviewer, we have significantly revised the introduction to emphasize this point and clarify the goals of this study. Specifically, we added several sentences in the introduction to establish the link between biomass production and temperature (second paragraph of the introduction on page 4):

"One of the primary environmental drivers of algal productivity is temperature. During mass cultivation in open ponds, algae are subject to fluctuations in temperature, which have dramatic effects on growth, impacting overall productivity and biomass composition⁶⁻⁸. As reported for several taxa^{9,10}, cold stress, either individually^{11,12} or in some combination¹³, reduces growth/productivity and can result in increased lipid that exceeds the effects of nitrogen starvation¹⁴ including unsaturated fatty acids^{11,12,15} that maintain membrane permeability and fluidity^{15,16}. However, the underlying mechanisms that control and regulate these and other physiological responses and adaptation to temperature stress are largely unknown. A comprehensive, systems level analysis of promising strains grown under industrially relevant suboptimal temperatures can help identify genetic targets and biochemical pathways that when modified, enhance stability and productivity of the algae during outdoor cultivation."

3. The author characterized the optimal and limiting growth conditions for NREL 46B-D3, e.g. salinity, N:P ratio. Light is also an important factor for algal cultivation but has never been mentioned in the paper.

We agree on the importance of light as a factor for algal cultivation. We now report the light intensity value, as noted in the response to this reviewer's comment #1. However, in this study we were focused on understanding the temperature specific response of NREL 46B-D3 and did not use light as a limiting factor in these studies.

4. The result that NREL 46B-D3 produces more biomass than other strains is very interesting. But the reason for that is not clear. Does NREL 46B-D3 grow faster and/or have higher photosynthetic rates (e.g. O₂ evolution rate) or higher tolerance to stresses than other strains?

We hypothesize that 46B-D3's capacity for photosynthetic biomass accumulation in stationary phase in part explains the observed biomass differential between the strains screened. Additionally, we hypothesize that 46B-D3's halotolerance confers an additional growth enhancement via higher stress tolerance, as noted in the manuscript text regarding the UTEX 393 comparison. We have updated the manuscript text to better explain the high biomass accumulation of NREL 46B-D3 on page 19:

"Thus, we hypothesize that the high biomass accumulation phenotype, relative to other strains evaluated during screening, is due in part to this high storage carbon accumulation phenotype post cell division."

Minor comments:

1. Fig 1a, are the NREL 46B-D3 cells always in the clumping status?

We imaged cells that were at a high density to capture multiple cells in one image, and thus appear to be clumping. Clumping is not a hallmark of this strain.

2. Does Fig1b have error bars? Why not including UTEX 393 in Fig 1b?

As noted in the figure legend this data is representative of screening data. In multiple screening analyses the same trend was always observed, NREL 46B-D3 produced more biomass than all other strains tested. UTEX 393 was not included in Figure 1b, as initial screening was carried out on a culture collection established at NREL, and UTEX 393 was not part of this culture collection.

3. Fig 1C, the data is presented as g/L, what was the cell density difference between these two strains? Did NREL 46B-D3 grow faster than UTEX 393 or does NREL 46B-D3 have more biomass per cell? Do you have OD750 curves for NREL 46B-D3 and UTEX 393 for comparison?

We agree with the reviewer that cell density would be an interesting comparison, however, we only measured endpoint ash-free dry weight, which we felt was the most pertinent metric for mass cultivation outdoor deployment.

4. Fig.4, why was there a drop of OD750 at 2day control cold?

Experiments for heat and cold perturbation were conducted separately (different points in time), each with a respective control. The “control-cold” OD at Day 2 vs the ‘control-hot’ OD on Day 2 is not significant and represents the natural biological variation in the algae.

5. Fig 7C is confusing. It is difficult to understand the first group and 2nd group mentioned in the paper.

We have modified the text in the Results section on page 13 to clarify the locations of the two groups in the co-expression network shown in Figure 7c, which now reads:

“To analyze the global patterns of expression, we constructed a co-expression network produced by the pairwise correlations between genes computed by Weighted Gene Co-expression Analysis (WGCNA), revealing two major groups of genes in the network structure (Figure 7c). The first group, which are the genes on the left side of the co-expression network, consists of genes that decreased in expression in the cold stress samples compared to control samples. The second group, which is the larger group of genes on the right side of the network, is mostly composed of genes that increased in expression in the cold stress samples, including a subset of genes that had lower expression in both stress conditions compared to control samples.”

6. Supplementary Fig. 6, it would be more elegant if putting the log2FC color bar by the side of the heatmap, but not on the top.

We updated the location of the color bar in Supplementary Fig. 6 as suggested.

7. “Thirty-five up-regulated and 46 down-regulated TFs were differentially expressed exclusively in cold stress samples.” Is this for Fig 7b? If so, the numbers don’t match the figure.

We thank the reviewer for pointing out the inconsistencies in the numbers. The number of up-regulated and down-regulated TFs should match with the numbers shown in the Venn diagram for Figure 7b, and we have corrected the numbers described in the text on page 15. We also now cite Figure 7b directly in the text.

8. “Changes in the expression of genes involved in fatty acid synthesis were only observed in the cold stress but not the heat stress, suggesting that fatty acid synthesis is only modulated in *Scenedesmus* sp. NREL 46B-D3 in response to cold.” I’m not sure if the statement is correct. It could also be because the heat stress was not strong enough to see the effects on the expression of genes involved in fatty acid synthesis.

We have modified the text in the Results section (page 16) to state that the fatty acid synthesis is modulated only in the cold “under conditions tested here”.

9. The authors mentioned “General amino acid accumulation is a mechanism to relieve oxidative stress”. Maybe have more discussion about this. What is the source of oxidative stress? What’s the function of amino acids in tolerance of oxidative stress?

We describe possible implications of amino acid accumulation (page 23), particularly arginine in response to oxidative stress most likely caused by cold stress and the subsequent generation of H₂O₂.

10. The strain selection condition is “temperature cycles from 21 to 32 °C, while lighting cycles from 0 to 965 μmol m⁻² s⁻¹.” Maybe add some details of these cycles, how long for 21 or 32°C?

We have added a reference to Arora et al. (*Biotechnology Advances*, 2019, <https://doi.org/10.1016/j.biotechadv.2018.04.005>) on page 25 that has additional details on these temperature and lighting cycles.

Did the authors use direct temperature change or gradual temperature change? The omics data under 33.2°C and 13°C may not explain the increased biomass accumulation in this strain.

Heat and cold samples were cultivated at the control temperature before being moved into their respective growth chambers. While we did not measure the rate at which the temperature changed, we can roughly estimate that cultures reached their new temperature within the first few hours and well before sampling began.

While we acknowledge that the omics data under these temperatures may not reveal why there is increased biomass accumulation, the omics data addresses the question about systems level changes of the transcriptional and metabolic response to temperature stress for this strain. The increased biomass accumulation is explained, in part, by the data presented in Figure 2, wherein significant biomass is accumulated post cell division.

REVIEWERS' COMMENTS:

Reviewer #1 (Remarks to the Author):

The authors have largely addressed my earlier comments satisfactorily.

My remaining main concern is about the approach for identifying a haplotype representation of the genome (page 29). It is important to note that the size of a genome assembly (i.e. the total assembled bases in an assembly, in this case 103.4 Mbp for the de-duplicated scaffolds) is not a good representation of the estimated (haploid/haplotype) genome size. Commonly in genome studies, the assessment of genome size (including %heterogeneity) is estimated using k-mers, e.g. implemented in GenomeScope2, and ploidy can be estimated using smudgeplot (<https://www.nature.com/articles/s41467-020-14998-3>). The haplotype representation of a genome may be generated using purge_haplotigs (<https://bmcbioinformatics.biomedcentral.com/articles/10.1186/s12859-018-2485-7>) that takes into consideration both read depth and sequence similarity to discern a haplotype representation. A brief justification as to why these established methods are not used would be great. Alternatively, I think it is important for the authors to (a) cite relevant support from the literature where available for their described approach that is based entirely on sequence similarity and lengths, (b) clarify "primary" versus "secondary" scaffolds, and (c) clarify that the assembly after removal of secondary scaffolds represents an approximate (or putative) haplotype genome assembly. The 130.4Mbp is the assembly size after you processed the assembly following your approach. It is not the estimated haplotype genome size.

Minor comment:

The "completeness of the genome annotation" (pages 7 and 30) is ambiguous and misleading. This phrase implies the comprehensiveness of the annotated genomic regions (both structurally and functionally), encompassing all features beyond protein-coding genes, i.e. to include repeats, various types of non-coding RNAs, UTRs, etc. – I believe this is not what you meant. You assessed genome completeness using the predicted proteins, which recovered 93.1% of complete core conserved Chlorophyta proteins in BUSCO (chlorophyta_odb10). Also, the instances of "chlorophyta_obd10" are typos, since the database is "chlorophyta_odb10".

Reviewer #2 (Remarks to the Author):

The revised manuscript addressed the comments in an acceptable way.

Reviewer #3 (Remarks to the Author):

The authors have addressed my comments well.

REVIEWERS' COMMENTS:

Reviewer #1 (Remarks to the Author):

The authors have largely addressed my earlier comments satisfactorily.

My remaining main concern is about the approach for identifying a haplotype representation of the genome (page 29). It is important to note that the size of a genome assembly (i.e the total assembled bases in an assembly, in this case 103.4 Mbp for the de-duplicated scaffolds) is not a good representation of the estimated (haploid/haplotype) genome size. Commonly in genome studies, the assessment of genome size (including %heterogeneity) is estimated using k-mers, e.g. implemented in GenomeScope2, and ploidy can be estimated using smudgeplot (<https://www.nature.com/articles/s41467-020-14998-3>). The haplotype representation of a genome may be generated using purge_haplotigs (<https://bmcbioinformatics.biomedcentral.com/articles/10.1186/s12859-018-2485-7>) that takes into consideration both read depth and sequence similarity to discern a haplotype representation. A brief justification as to why these established methods are not used would be great. Alternatively, I think it is important for the authors to (a) cite relevant support from the literature where available for their described approach that is based entirely on sequence similarity and lengths, (b) clarify “primary” versus “secondary” scaffolds, and (c) clarify that the assembly after removal of secondary scaffolds represents an approximate (or putative) haplotype genome assembly. The 130.4Mbp is the assembly size after you processed the assembly following your approach. It is not the estimated haplotype genome size.

While genome size can be estimated using k-mer frequencies with tools like GenomeScope2 and smudgeplot for low error short read datasets, these approaches are not designed to be used on long read datasets such as the PacBio RSII dataset produced for the genome of *Scenedesmus* sp. NREL 46B-D3. The authors of these tools state that “GenomeScope and Smudgeplot only support low error short read sequencing. Future work remains to extend these techniques for single molecule sequencing with high error rates that currently prevent k-mer based analysis.” (Ranallo-Benavidez, T.R., Jaron, K.S. and Schatz, M.C. Nature Communications 11, 1432 (2020). <https://doi.org/10.1038/s41467-020-14998-3>). On the other hand, methods similar to our own approach to remove duplicate scaffolds based on genome-wide alignments are applied in other annotation pipelines, such as funannotate (Palmer, J. and Stajich, J. 2020. nextgenusfs/funannotate: funannotate v1.8.1. Zenodo. <https://doi.org/10.5281/zenodo.4054262>).

Regarding the distinction between primary and secondary scaffolds, we refer the reviewer to the description on page 29:

“During the annotation process, secondary scaffolds were detected by performing an all-against-all BLAT alignment⁷³. If a pair of scaffolds were aligned with >50% coverage and >95% sequence identity, the longer scaffold was labeled as the primary scaffold and the shorter scaffold was labeled as the secondary scaffold.”

Although we believe that our approach provides a good estimation of the haplotype assembly size, we used the `purge_haplotigs` pipeline suggested by the reviewer as an alternative approach to generate a haplotype representation based on read depth coverage. The `purge_haplotigs` pipeline generated a haplotype representation of 103.5 Mbp, largely consistent with our approach, within 100Kbp of our estimate of 103.4 Mbp. Of the 780 scaffolds in the `purge_haplotigs` assembly, 618 scaffolds were also defined as primary scaffolds in our approach accounting for about 98% of the sequence in the assembly. The haplotype generated by our pipeline includes fewer but longer scaffolds (763 primary scaffolds) as compared to `purge_haplotigs` because we assign the smaller of an allelic scaffold pair to the secondary scaffold set. Only for or a small fraction of short (11.9Kbp on average and median of 5.7kb) scaffolds, there are discrepancies between the assignment of primary scaffold or secondary scaffold ("haplotig"). Given the similarities between haplotype representations produced by our pipeline and `purge_haplotigs`, we believe that our approach is appropriate for estimating the haploid assembly size. We revised the language in the text in the Methods to clarify that the genome assembly size is an approximate estimate (page 29):

“Excluding the secondary scaffolds from the genome assembly, the assembly size of the single haplotype is approximated to be 103.4 Mbp.”

Minor comment:

The “completeness of the genome annotation” (pages 7 and 30) is ambiguous and misleading. This phrase implies the comprehensiveness of the annotated genomic regions (both structurally and functionally), encompassing all features beyond protein-coding genes, i.e. to include repeats, various types of non-coding RNAs, UTRs, etc. – I believe this is not what you meant. You assessed genome completeness using the predicted proteins, which recovered 93.1% of complete core conserved Chlorophyta proteins in BUSCO (`chlorophyta_odb10`). Also, the instances of “`chlorophyta_obd10`” are typos, since the database is “`chrolophyta_obd10`”.

Regarding the reviewer’s comment on assessment of completeness, we edited the text to emphasize the point that the completeness evaluated using BUSCO is estimated for only the protein-coding genes in the annotation. We’ve also edited the typos.

On page 7:

“The completeness of the predicted protein-coding gene set was estimated to be 93.1% complete based on Chlorophyta BUSCO families (chlorophyta_odb10; 11-20-19)^{26,27} (Supplementary Figure 3).”

On page 30:

“The completeness of the predicted protein-coding gene set was evaluated using BUSCO v 4.0.5 in protein mode on the Chlorophyta ortholog dataset (chlorophyta_odb10; 11-20-19)⁸⁶.”

Reviewer #2 (Remarks to the Author):

The revised manuscript addressed the comments in an acceptable way.

Reviewer #3 (Remarks to the Author):

The authors have addressed my comments well.

We thank all the reviewers for their comments.